# Agnostic Learning of a Single Neuron with Gradient Descent

**Spencer Frei**
Department of Statistics
University of California, Los Angeles
Los Angeles, CA 90095, USA
spencerfrei@ucla.edu

**Yuan Cao**
Department of Computer Science
University of California, Los Angeles
Los Angeles, CA 90095, USA
yuancao@cs.ucla.edu

**Quanquan Gu**
Department of Computer Science
University of California, Los Angeles
Los Angeles, CA 90095, USA
qgu@cs.ucla.edu

## Abstract

We consider the problem of learning the best-fitting single neuron as measured by the expected square loss $\mathbb{E}_{(x,y)\sim\mathcal{D}}[(\sigma(w^\top x) - y)^2]$ over some unknown joint distribution $\mathcal{D}$ by using gradient descent to minimize the empirical risk induced by a set of i.i.d. samples $S \sim \mathcal{D}^n$. The activation function $\sigma$ is an arbitrary Lipschitz and non-decreasing function, making the optimization problem nonconvex and nonsmooth in general, and covers typical neural network activation functions and inverse link functions in the generalized linear model setting. In the agnostic PAC learning setting, where no assumption on the relationship between the labels $y$ and the input $x$ is made, if the optimal population risk is $\mathsf{OPT}$, we show that gradient descent achieves population risk $O(\mathsf{OPT}) + \varepsilon$ in polynomial time and sample complexity when $\sigma$ is strictly increasing. For the ReLU activation, our population risk guarantee is $O(\mathsf{OPT}^{1/2}) + \varepsilon$. When labels take the form $y = \sigma(v^\top x) + \xi$ for zero-mean sub-Gaussian noise $\xi$, we show that the population risk guarantees for gradient descent improve to $\mathsf{OPT} + \varepsilon$. Our sample complexity and runtime guarantees are (almost) dimension independent, and when $\sigma$ is strictly increasing, require no distributional assumptions beyond boundedness. For ReLU, we show the same results under a nondegeneracy assumption for the marginal distribution of the input.

## 1   Introduction

We study learning the best possible single neuron that captures the relationship between the input $x \in \mathbb{R}^d$ and the output label $y \in \mathbb{R}$ as measured by the expected square loss over some unknown but fixed distribution $(x, y) \sim \mathcal{D}$. In particular, for a given activation function $\sigma : \mathbb{R} \to \mathbb{R}$, we define the population risk $F(w)$ associated with a set of weights $w$ as

$$F(w) := (1/2)\mathbb{E}_{(x,y)\sim\mathcal{D}}\left[\left(\sigma(w^\top x) - y\right)^2\right]. \tag{1.1}$$

The activation function is assumed to be non-decreasing and Lipschitz, and includes nearly all activation functions used in neural networks such as the rectified linear unit (ReLU), sigmoid, $\tanh$, and so on. In the agnostic PAC learning setting [23], no structural assumption is made regarding the relationship of the input and the label, and so the best-fitting neuron could, in the worst case, have nontrivial population risk. Concretely, if we denote

$$v := \operatorname{argmin}_{\|w\|_2 \leq 1} F(w), \quad \mathsf{OPT} := F(v), \tag{1.2}$$

then the goal of a learning algorithm is to (efficiently) return weights $w$ such that the population risk $F(w)$ is close to the best possible risk $\mathsf{OPT}$. The agnostic learning framework stands in contrast to the *realizable* PAC learning setting, where one assumes $\mathsf{OPT} = 0$, so that there exists some $v$ such that the labels are given by $y = \sigma(v^\top x)$.

The learning algorithm we consider in this paper is empirical risk minimization using vanilla gradient descent. We assume we have access to a set of i.i.d. samples $\{(x_i, y_i)\}_{i=1}^n \sim \mathcal{D}^n$, and we run gradient descent with a fixed step size on the empirical risk $\widehat{F}(w) = (1/2n) \sum_{i=1}^n (\sigma(w^\top x_i) - y_i)^2$. A number of early neural network studies pointed out that the landscape of the empirical risk of a single neuron has unfavorable properties, such as a large number of spurious local minima [6, 3], and led researchers to instead study gradient descent on a convex surrogate loss [16, 17]. Despite this, we are able to show that gradient descent on the empirical risk itself finds weights that not only have small empirical risk but small population risk as well.

Surprisingly little is known about neural networks trained by minimizing the empirical risk with gradient descent in the agnostic PAC learning setting. We are aware of two works [2, 1] in the *improper* agnostic learning setting, where the goal is to return a hypothesis $h \in \mathcal{H}$ that achieves population risk close to $\widehat{\mathsf{OPT}}$, where $\widehat{\mathsf{OPT}}$ is the smallest possible population risk achieved by a different set of hypotheses $\widehat{\mathcal{H}}$. Another work considered the random features setting where only the final layer of the network is trained and the marginal distribution over $x$ is uniform on the unit sphere [34]. But none of these address the simplest possible neural network: that of a single neuron $x \mapsto \sigma(w^\top x)$. We believe a full characterization of what we can (or cannot) guarantee for gradient descent in the single neuron setting will help us understand what is possible in the more complicated deep neural network setting. Indeed, two of the most common hurdles in the analysis of deep neural networks trained by gradient descent—nonconvexity and nonsmoothness—are also present in the case of the single neuron. We hope that our analysis in this relatively simple setup will be suggestive of what is possible in more complicated neural network models.

Our main contributions can be summarized as follows.

1) **Agnostic setting** (Theorem 3.3). Without any assumptions on the relationship between $y$ and $x$, and assuming only boundedness of the marginal distributions of $x$ and $y$, we show that for any $\varepsilon > 0$, gradient descent finds a point $w_t$ with population risk $O(\mathsf{OPT}) + \varepsilon$ with sample complexity $O(\varepsilon^{-2})$ and runtime $O(\varepsilon^{-1})$ when $\sigma(\cdot)$ is strictly increasing and Lipschitz. When $\sigma$ is ReLU, we obtain a population risk guarantee of $O(\mathsf{OPT}^{1/2}) + \varepsilon$ with sample complexity $O(\varepsilon^{-4})$ and runtime $O(\varepsilon^{-2})$ when the marginal distribution of $x$ satisfies a nondegeneracy condition (Assumption 3.2). The sample and runtime complexities are independent of the input dimension for both strictly increasing activations and ReLU.

2) **Noisy teacher network setting** (Theorem 4.1). When $y = \sigma(v^\top x) + \xi$, where $\xi|x$ is zero-mean and sub-Gaussian (and possibly dependent on $x$), we demonstrate that gradient descent finds $w_t$ satisfying $F(w_t) \leq \mathsf{OPT} + \varepsilon$ for activation functions that are strictly increasing and Lipschitz assuming only boundedness of the marginal distribution over $x$. The same result holds for ReLU under a marginal spread assumption (Assumption 3.2). The runtime and sample complexities are of order $\tilde{O}(\varepsilon^{-2})$, with a logarithmic dependence on the input dimension. When the noise is bounded, our guarantees are dimension independent. If we further know $\xi \equiv 0$, i.e. the learning problem is in the realizable rather than agnostic setting, we can improve the runtime and sample complexity guarantees from $O(\varepsilon^{-2})$ to $O(\varepsilon^{-1})$ by using online stochastic gradient descent (Theorem E.1).

## 2   Related work

Below, we provide a high-level summary of related work in the agnostic learning and teacher network settings. Detailed comparisons with the most related works will appear after we present our main theorems in Sections 3 and 4. In Appendix A, we provide tables that describe the assumptions and complexity guarantees of our work in comparison to related results.

**Agnostic learning:** The simplest version of the agnostic regression problem is to find a hypothesis that matches the performance of the best *linear* predictor. In our setting, this corresponds to $\sigma$ being the identity function. This problem is completely characterized: Shamir [29] showed that any algorithm that returns a linear predictor $v$ has risk $\mathsf{OPT} + \Omega(\varepsilon^{-2} \wedge d\varepsilon^{-1})$ when the labels satisfy

$|y| \leq 1$ and the marginal distribution over $x$ is supported on the unit ball, matching upper bounds proved by Srebro et al. [32] using mirror descent.

When $\sigma$ is not the identity, related works are scarce. Goel et al. [12] studied agnostic learning of the ReLU on distributions supported on the unit sphere but had runtime and sample complexity exponential in $\varepsilon^{-1}$. In another work on learning a single ReLU, Goel et al. [14] showed that learning up to risk $\mathsf{OPT} + \varepsilon$ in polynomial time is as hard as the problem of learning sparse parities with noise, long believed to be computationally intractable. Additionally, they provided an approximation algorithm that could learn up to $O(\mathsf{OPT}^{2/3}) + \varepsilon$ risk in $\mathrm{poly}(d, \varepsilon^{-1})$ time and sample complexity when the marginal distribution over $x$ is a standard Gaussian. In a related but incomparable set of results in the improper agnostic learning setting, Allen-Zhu et al. [2] and Allen-Zhu and Li [1] showed that multilayer ReLU networks trained by gradient descent can match the population risk achieved by multilayer networks with smooth activation functions. Vempala and Wilmes [34] studied agnostic learning of a one-hidden-layer neural network when the first layer is fixed at its (random) initial values and the second layer is trained. A very recent work by Diakonikolas et al. [8] showed that population risk $O(\mathsf{OPT}) + \varepsilon$ can be achieved for the single ReLU neuron by appealing to gradient descent on a convex surrogate for the empirical risk.

**Teacher network:** The literature refers to the case of $y = \sigma(v^\top x) + \xi$ for some possible zero mean noise $\xi$ variously as the "noisy teacher network" or "generalized linear model" (GLM) setting, and is related to the probabilistic concepts model [22]. In the GLM setting, $\sigma$ plays the role of the inverse link function; in the case of logistic regression, $\sigma$ is the sigmoid function.

The results in the teacher network setting can be broadly characterized by (1) whether they cover arbitrary distributions over $x$ and (2) the presence of noise (or lackthereof). The GLMTron algorithm proposed by Kakade et al. [20], itself a modification of the Isotron algorithm of Kalai and Sastry [21], is known to learn a noisy teacher network up to risk $\mathsf{OPT} + \varepsilon$ for any Lipschitz and non-decreasing $\sigma$ and any distribution with bounded marginals over $x$. Mei et al. [25] showed that gradient descent learns the noisy teacher network under a smoothness assumption on the activation function for a large class of distributions. Foster et al. [11] provided a meta-algorithm for translating $\varepsilon$-stationary points of the empirical risk to points of small population risk in the noisy teacher network setting. A recent work by Mukherjee and Muthukumar [26] develops a modified SGD algorithm for learning a ReLU with bounded adversarial noise on distributions where the input is bounded.

Of course, any guarantee that holds for a neural network with a single fully connected hidden layer of arbitrary width holds for the single neuron, so in this sense our work can be connected to a larger body of work on the analysis of gradient descent used for learning neural networks. The majority of such works are restricted to particular input distributions, whether it is Gaussian or uniform distributions [30, 33, 31, 36, 13, 7]. Du et al. [10] showed that in the noiseless (a.k.a. realizable) setting, a single neuron can be learned with SGD if the input distribution satisfies a certain subspace eigenvalue property. Yehudai and Shamir [35] studied the properties of learning a single neuron for a variety of increasing and Lipschitz activation functions using gradient descent, as we do in this paper, although their analysis was restricted to the noiseless setting.

## 3 Agnostic learning setting

We begin our analysis by assuming there is no *a priori* relationship between $x$ and $y$, so the population risk $\mathsf{OPT}$ of the population risk minimizer $v$ defined in (1.2) may, in general, be a large quantity. If $\mathsf{OPT} = 0$, then $\sigma(v^\top x) = y$ a.s. and the problem is in the realizable PAC learning setting. In this case, we can use a modified proof technique to get stronger guarantees for the population risk; see Appendix E for the complete theorems and proofs in this setting. We will thus assume without loss of generality that $0 < \mathsf{OPT} \leq 1$.

The gradient descent method we use in this paper is as follows. We assume we have a training sample $\{(x_i, y_i)\}_{i=1}^{n} \overset{\text{i.i.d.}}{\sim} \mathcal{D}^n$, and define the empirical risk for weight $w$ by

$$\widehat{F}(w) = (1/2n) \sum_{i=1}^{n} (\sigma(w^\top x_i) - y_i)^2.$$

We perform full-batch gradient updates on the empirical risk using a fixed step size $\eta$,

$$w_{t+1} = w_t - \eta \nabla \widehat{F}(w_t) = w_t - (\eta/n) \sum_{i=1}^{n} (\sigma(w_t^\top x_i) - y_i)\sigma'(w_t^\top x_i)x_i, \tag{3.1}$$

where $\sigma'(\cdot)$ is the derivative of $\sigma(\cdot)$. If $\sigma$ is not differentiable at a point $z$, we will use its subderivative.

We begin by describing one set of activation functions under consideration in this paper.

**Assumption 3.1.** (a) $\sigma$ is continuous, non-decreasing, and differentiable almost everywhere.

(b) For any $\rho > 0$, there exists $\gamma > 0$ such that $\inf_{|z| \leq \rho} \sigma'(z) \geq \gamma > 0$. If $\sigma$ is not differentiable at $z \in [-\rho, \rho]$, assume that every subderivative $g$ on the interval satisfies $g(z) \geq \gamma$.

(c) $\sigma$ is $L$-Lipschitz, i.e. $|\sigma(z_1) - \sigma(z_2)| \leq L|z_1 - z_2|$ for all $z_1, z_2$.

We note that if $\sigma$ is strictly increasing and continuous, then $\sigma$ satisfies Assumption 3.1(b) since its derivative is never zero. In particular, the assumption covers the typical activation functions in neural networks like leaky ReLU, softplus, sigmoid, tanh, etc., but excludes ReLU. Yehudai and Shamir [35] recently showed that when $\sigma$ is ReLU, there exists a distribution $\mathcal{D}$ supported on the unit ball and unit length target neuron $v$ such that *even in the realizable case* of $y = \sigma(v^\top x)$, if the weights are initialized randomly using a product distribution, there exists a constant $c_0$ such that with high probability, $F(w_t) \geq c_0 > 0$ throughout the trajectory of gradient descent. This suggests that gradient-based methods for learning ReLUs are likely to fail without additional assumptions. Because of this, they introduced the following marginal spread assumption to handle the learning of ReLU.

**Assumption 3.2.** There exist constants $\alpha, \beta > 0$ such that the following holds. For any $w \neq u$, denote by $\mathcal{D}_{w,u}$ the marginal distribution of $\mathcal{D}$ on $\mathrm{span}(w, u)$, viewed as a distribution over $\mathbb{R}^2$, and let $p_{w,u}$ be its density function. Then $\inf_{z \in \mathbb{R}^2 : \|z\| \leq \alpha} p_{w,u}(z) \geq \beta$.

This assumption covers, for instance, log-concave distributions like the Gaussian and uniform distribution with $\alpha, \beta = O(1)$ [24]. We note that a similar assumption was used in recent work on learning halfspaces with Massart noise [9]. We will use this assumption for all of our results when $\sigma$ is ReLU. Additionally, although the ReLU is not differentiable at the origin, we will denote by $\sigma'(0)$ its subderivative, with the convention that $\sigma'(0) = 1$. Such a convention is consistent with the implementation of ReLUs in modern deep learning software packages.

With the above in hand, we can describe our main theorem.

**Theorem 3.3.** Suppose the marginals of $\mathcal{D}$ satisfy $\|x\|_2 \leq B_X$ a.s. and $|y| \leq B_Y$ a.s. Let $a := (|\sigma(B_X)| + B_Y)^2$. When $\sigma$ satisfies Assumption 3.1, let $\gamma > 0$ be the constant corresponding to $\rho = 2B_X$ and fix a step size $\eta \leq (1/8)\gamma L^{-3} B_X^{-2}$. For any $\delta > 0$, with probability at least $1 - \delta$, gradient descent initialized at the origin and run for $T = \lceil \eta^{-1} \gamma^{-1} L^{-1} B_X^{-1} [\mathsf{OPT} + an^{-1/2} \log^{1/2}(4/\delta)]^{-1} \rceil$ iterations finds weights $w_t, t < T$, such that

$$F(w_t) \leq C_1 \mathsf{OPT} + C_2 n^{-1/2}, \tag{3.2}$$

where $C_1 = 12\gamma^{-3}L^3 + 2$ and $C_2 = O(L^3 B_X^2 \sqrt{\log(1/\delta)} + C_1 a \sqrt{\log(1/\delta)})$.

When $\sigma$ is ReLU, further assume that $\mathcal{D}_x$ satisfies Assumption 3.2 for constants $\alpha, \beta > 0$, and let $\nu = \alpha^4 \beta / 8\sqrt{2}$. Fix a step size $\eta \leq (1/4) B_X^{-2}$. For any $\delta > 0$, with probability at least $1 - \delta$, gradient descent initialized at the origin and run for $T = \lceil \eta^{-1} B_X^{-1} [\mathsf{OPT} + an^{-1/2} \log^{1/2}(4/\delta)]^{-1/2} \rceil$ iterations finds a point $w_t$ such that

$$F(w_t) \leq C_1 \mathsf{OPT}^{1/2} + C_2 n^{-1/4} + C_3 n^{-1/2}, \tag{3.3}$$

where $C_1 = O(B_X \nu^{-1})$, $C_2 = O(C_1 a^{1/2} \log^{1/4}(1/\delta))$, and $C_3 = O(B_X^2 \nu^{-1} \log^{1/2}(1/\delta))$.

We remind the reader that the optimization problem for the empirical risk is highly nonconvex [3] and thus any guarantee for the empirical risk, let alone the population risk, is nontrivial. This makes us unsure if the suboptimal guarantee of $O(\mathsf{OPT}^{1/2})$ for ReLU is an artifact of our analysis or a necessary consequence of nonconvexity.

In comparison to recent work, Goel et al. [14] considered the agnostic setting for the ReLU activation when the marginal distribution over $x$ is a standard Gaussian and showed that learning up to risk $\mathsf{OPT} + \varepsilon$ is as hard as learning sparse parities with noise. By using an approximation algorithm of Awasthi et al. [4], they were able to show that one can learn up to $O(\mathsf{OPT}^{2/3}) + \varepsilon$ with $O(\mathrm{poly}(d, \varepsilon^{-1}))$ runtime and sample complexity. In a very recent work, Diakonikolas et al. [8] improved the population risk guarantee for the ReLU to $O(\mathsf{OPT}) + \varepsilon$ when the features are sampled from an isotropic log-concave distribution by analyzing gradient descent on a convex surrogate loss. Projected gradient descent on this surrogate loss produces the weight updates of the GLMTron algorithm of Kakade et al. [20]. Using the solution found by gradient descent on the surrogate loss, they proposed an improper

learning algorithm that improves the population risk guarantee from $O(\mathsf{OPT}) + \varepsilon$ to $(1 + \delta)\mathsf{OPT} + \varepsilon$ for any $\delta > 0$.

By contrast, we show that gradient descent on the empirical risk learns up to a population risk of $O(\mathsf{OPT}) + \varepsilon$ for *any* joint distribution with bounded marginals when $\sigma$ is strictly increasing and Lipschitz, even though the optimization problem is nonconvex. In the case of ReLU, our guarantee holds for the class of bounded distributions over $x$ that satisfy the marginal spread condition of Assumption 3.2 and hence covers (bounded) log-concave distributions, although the guarantee is $O(\mathsf{OPT}^{1/2})$ in this case. For all activation functions we consider, the runtime and sample complexity guarantees do not have (explicit) dependence on the dimension.[1] Moreover, we shall see in the next section that if the data is known to come from a noisy teacher network, the guarantees of gradient descent improve to $\mathsf{OPT} + \varepsilon$ for both strictly increasing activations and ReLU.

In the remainder of this section we will prove Theorem 3.3. Our proof relies upon the following auxiliary errors for the true risk $F$:

$$G(w) := (1/2)\mathbb{E}_{(x,y)\sim\mathcal{D}}\left[\left(\sigma(w^\top x) - \sigma(v^\top x)\right)^2\right],$$

$$H(w) := (1/2)\mathbb{E}_{(x,y)\sim\mathcal{D}}\left[\left(\sigma(w^\top x) - \sigma(v^\top x)\right)^2 \sigma'(w^\top x)\right]. \tag{3.4}$$

We will denote the corresponding empirical risks by $\widehat{G}(w)$ and $\widehat{H}(w)$. We first note that $G$ trivially upper bounds $F$: this follows by a simple application of Young's inequality and, when $\mathbb{E}[y|x] = \sigma(v^\top x)$, by using iterated expectations.

**Claim 3.4.** For any joint distribution $\mathcal{D}$, for any vector $u$, and any continuous activation function $\sigma$, $F(u) \leq 2G(u) + 2F(v)$. If additionally we know that $\mathbb{E}[y|x] = \sigma(v^\top x)$, we have $F(u) = G(u) + F(v)$.

This claim shows that in order to show the population risk is small, it suffices to show that $G$ is small. It is easy to see that if $\inf_{z\in\mathbb{R}} \sigma'(z) \geq \gamma > 0$, then $H(w) \leq \varepsilon$ implies $G(w) \leq \gamma^{-1}\varepsilon$, but the only typical activation function that satisfies this condition is the leaky ReLU. Fortunately, when $\sigma$ satisfies Assumption 3.1, or when $\sigma$ is ReLU and $\mathcal{D}$ satisfies Assumption 3.2, Lemma 3.5 below shows that $H$ is still an upper bound for $G$. The proof is deferred to Appendix B.

**Lemma 3.5.** If $\sigma$ satisfies Assumption 3.1, $\|x\|_2 \leq B$ a.s., and $\|w\|_2 \leq W$, then for $\gamma$ corresponding to $\rho = WB$, $H(w) \leq \varepsilon$ implies $G(w) \leq \gamma^{-1}\varepsilon$. If $\sigma$ is ReLU and $\mathcal{D}$ satisfies Assumption 3.2 for some constants $\alpha, \beta > 0$, and if for some $\varepsilon > 0$ the bound $H(w) \leq \beta\alpha^4\varepsilon/8\sqrt{2}$ holds, then $\|w - v\|_2 \leq 1$ implies $G(w) \leq \varepsilon$.

Claim 3.4 and Lemma 3.5 together imply that if gradient descent finds a point with auxiliary error $H(w_t) \leq O(\mathsf{OPT}^\alpha)$ for some $\alpha \leq 1$, then gradient descent achieves population risk $O(\mathsf{OPT}^\alpha)$. In the remainder of this section, we will show that this is indeed the case. In Section 3.1, we first consider activations satisfying Assumption 3.1, for which we are able to show $H(w_t) \leq O(\mathsf{OPT})$. In Section 3.2, we show $H(w_t) \leq O(\mathsf{OPT}^{1/2})$ for the ReLU.

## 3.1 Strictly increasing activations

In Lemma 3.6 below, we show that $\widehat{H}(w_t)$ is a natural quantity of the gradient descent algorithm that in a sense tells us how good a direction the gradient is pointing at time $t$, and that $\widehat{H}(w_t)$ can be as small as $O(\widehat{F}(v))$. Our proof technique is similar to that of Kakade et al. [20], who studied the GLMTron algorithm in the (non-agnostic) noisy teacher network setup.

**Lemma 3.6.** Suppose that $\|x\|_2 \leq B_X$ a.s. under $\mathcal{D}_x$. Suppose $\sigma$ satisfies Assumption 3.1, and let $\gamma$ be the constant corresponding to $\rho = 2B_X$. Assume $\widehat{F}(v) > 0$. Gradient descent with fixed step size $\eta \leq (1/8)\gamma L^{-3}B_X^{-2}$ initialized at $w_0 = 0$ finds weights $w_t$ satisfying $\widehat{H}(w_t) \leq 6L^3\gamma^{-2}\widehat{F}(v)$ within $T = \lceil\eta^{-1}\gamma^{-1}L^{-1}B_X^{-1}\widehat{F}(v)^{-1}\rceil$ iterations, with $\|w_t - v\|_2 \leq 1$ for each $t = 0, \ldots, T - 1$.

Before beginning the proof, we first note the following fact, which allows us to connect terms that appear in the gradient to the square loss.

**Fact 3.7.** If $\sigma$ is strictly increasing on an interval $[a, b]$ with $\sigma'(z) \geq \gamma > 0$ for all $z \in [a, b]$, and if $z_1, z_2 \in [a, b]$, then, it holds that

$$\gamma(z_1 - z_2)^2 \leq (\sigma(z_1) - \sigma(z_2))(z_1 - z_2). \tag{3.5}$$

*Proof of Lemma 3.6.* The proof comes from the following induction statement. We claim that for every $t \in \mathbb{N}$, either (a) $\widehat{H}(w_\tau) \leq 6L^3\gamma^{-2}\widehat{F}(v)$ for some $\tau < t$, or (b) $\|w_t - v\|_2^2 \leq \|w_{t-1} - v\|_2^2 - \eta L\widehat{F}(v)$ holds. If this claim is true, then until gradient descent finds a point where $\widehat{H}(w_t) \leq 6L^3\gamma^{-2}\widehat{F}(v)$, the squared distance $\|w_t - v\|_2^2$ decreases by $\eta L\widehat{F}(v)$ at every iteration. Since $\|w_0 - v\|_2^2 = 1$, this means there can be at most $1/(\eta L\widehat{F}(v)) = \eta^{-1}L^{-1}\widehat{F}(v)^{-1}$ iterations until we reach $\widehat{H}(w_t) \leq 6L^3\gamma^{-2}\widehat{F}(v)$.

So let us now suppose the induction hypothesis holds for $t$, and consider the case $t + 1$. If (a) holds, then we are done. So now consider the case that for every $\tau \leq t$, we have $\widehat{H}(w_\tau) > 6L^3\gamma^{-2}\widehat{F}(v)$. Since (a) does not hold, $\|w_\tau - v\|_2^2 \leq \|w_{\tau-1} - v\|_2^2 - \eta L\widehat{F}(v)$ holds for each $\tau = 1, \ldots, t$, and so $\|w_0 - v\|_2 = 1$ implies

$$\|w_\tau - v\|_2 \leq 1 \ \forall \tau \leq t. \tag{3.6}$$

In particular, $\|w_\tau\|_2 \leq 1 + \|v\|_2 \leq 2$ holds for all $\tau \leq t$. By Cauchy–Schwarz, this implies $|w_\tau^\top x| \vee |v^\top x| \leq 2B_X$ a.s. By defining $\rho = 2B_X$ and letting $\gamma$ be the constant from Assumption 3.1, this implies $\sigma'(z) \geq \gamma > 0$ for all $|z| \leq 2B_X$. Fact 3.7 therefore implies

$$\sigma'(w_\tau^\top x) \geq \gamma > 0 \quad \text{and} \quad (\sigma(w_\tau^\top x) - \sigma(v^\top x)) \cdot (w_\tau^\top x - v^\top x) \geq \gamma(w_\tau^\top x - v^\top x)^2 \quad \forall \tau \leq t. \tag{3.7}$$

We proceed with the proof by demonstrating an appropriate lower bound for the quantity

$$\|w_t - v\|_2^2 - \|w_{t+1} - v\|_2^2 = 2\eta\left\langle \nabla\widehat{F}(w_t), w_t - v \right\rangle - \eta^2\left\|\nabla\widehat{F}(w_t)\right\|_2^2.$$

We begin with the inner product term. We have

$$
\begin{aligned}
\left\langle \nabla\widehat{F}(w_t), w_t - v \right\rangle &= (1/n)\sum_{i=1}^n \left(\sigma(w_t^\top x_i) - \sigma(v^\top x_i)\right)\sigma'(w_t^\top x_i)(w_t^\top x_i - v^\top x_i) \\
&\quad + (1/n)\sum_{i=1}^n \left(\sigma(v^\top x_i) - y_i\right)\gamma^{-1/2} \cdot \gamma^{1/2}\sigma'(w_t^\top x_i)(w_t^\top x_i - v^\top x_i) \\
&\geq (\gamma/n)\sum_{i=1}^n \left(w_t^\top x_i - v^\top x_i\right)^2\sigma'(w_t^\top x_i) \\
&\quad - \frac{\gamma^{-1}}{2n}\sum_{i=1}^n \left(\sigma(v^\top x_i) - y_i\right)^2\sigma'(w_t^\top x_i) - \frac{\gamma}{2n}\sum_{i=1}^n \left(w_t^\top x_i - v^\top x_i\right)^2\sigma'(w_t^\top x_i) \\
&\geq \frac{\gamma}{2}\sum_{i=1}^n (w_t^\top x_i - v^\top x_i)^2\sigma'(w_t^\top x_i) - L\gamma^{-1}\widehat{F}(v) \\
&\geq \gamma L^{-2}\widehat{H}(w_t) - L\gamma^{-1}\widehat{F}(v). \tag{3.8}
\end{aligned}
$$

In the first inequality we used (3.7) for the first term and Young's inequality for the second (and that $\sigma' \geq 0$). For the final two inequalities, we use that $\sigma$ is $L$-Lipschitz.

For the gradient upper bound,

$$
\begin{aligned}
\left\|\nabla\widehat{F}(w)\right\|^2 &\leq 2\left\|\frac{1}{n}\sum_{i=1}^n(\sigma(w^\top x_i) - \sigma(v^\top x_i))\sigma'(w^\top x_i)x_i\right\|^2 \\
&\quad + 2\left\|\frac{1}{n}\sum_{i=1}^n(\sigma(v^\top x_i) - y_i)\sigma'(w^\top x_i)x_i\right\|^2 \\
&\leq \frac{2}{n}\sum_{i=1}^n(\sigma(w^\top x_i) - \sigma(v^\top x_i))^2\sigma'(w^\top x_i)^2\|x_i\|_2^2
\end{aligned}
$$

$$+ \frac{2}{n} \sum_{i=1}^{n} (\sigma(v^\top x_i) - y_i)^2 \sigma'(w^\top x_i)^2 \|x_i\|_2^2$$

$$\leq \frac{2LB_X^2}{n} \sum_{i=1}^{n} (\sigma(w^\top x_i) - \sigma(v^\top x_i))^2 \sigma'(w^\top x_i) + 4L^2 B_X^2 \widehat{F}(v)$$

$$= 4LB_X^2 \widehat{H}(w) + 4L^2 B_X^2 \widehat{F}(v). \tag{3.9}$$

The first inequality is due to Young's inequality, and the second is due to Jensen's inequality. The last inequality holds because $\sigma$ is $L$-Lipschitz and $\|x\|_2 \leq B_X$ a.s. Putting (3.8) and (3.9) together and taking $\eta \leq (1/8)L^{-3} B_X^{-2} \gamma$,

$$\|w_t - v\|^2 - \|w_{t+1} - v\|^2 \geq 2\eta(\gamma L^{-2} \widehat{H}(w_t) - L\gamma^{-1} \widehat{F}(v)) - 4\eta^2 (LB_X^2 \widehat{H}(w_t) + L^2 B_X^2 \widehat{F}(v))$$

$$\geq 2\eta \left( {}^1\!/{}_2 \cdot \gamma L^{-2} \widehat{H}(w_t) - {}^5\!/{}_2 \cdot L\gamma^{-1} \widehat{F}(v) \right)$$

$$\geq \eta\gamma L\widehat{F}(v).$$

The last inequality uses the induction assumption that $\widehat{H}(w_t) \geq 6L^3 \gamma^{-2} \widehat{F}(v)$, completing the proof. □

Since the auxiliary error $\widehat{H}(w)$ is controlled by $\widehat{F}(v)$, we need to bound $\widehat{F}(v)$. When the marginals of $\mathcal{D}$ are bounded, Lemma 3.8 below shows that $\widehat{F}(v)$ concentrates around $F(v) = \mathsf{OPT}$ at rate $n^{-1/2}$ by Hoeffding's inequality; for completeness, the proof is given in Appendix F.

**Lemma 3.8.** If $\|x\|_2 \leq B_X$ and $|y| \leq B_Y$ a.s. under $\mathcal{D}_x$ and $\mathcal{D}_y$ respectively, and if $\sigma$ is non-decreasing, then for $a := \left( |\sigma(B_X)| + B_Y \right)^2$ and $\|v\|_2 \leq 1$, we have with probability at least $1 - \delta$,

$$|\widehat{F}(v) - \mathsf{OPT}| \leq 3a\sqrt{n^{-1} \log(2/\delta)}.$$

The final ingredient to the proof is translating the bounds for the empirical risk to one for the population risk. Since $\mathcal{D}_x$ is bounded and we showed in Lemma 3.6 that $\|w_t - v\|_2 \leq 1$ throughout the gradient descent trajectory, we can use standard properties of Rademacher complexity to get it done. The proof for Lemma 3.9 can be found in Appendix F.

**Lemma 3.9.** Suppose $\sigma$ is $L$-Lipschitz and $\|x\|_2 \leq B_X$ a.s. Denote $\ell(w; x) := (1/2)\left( \sigma(w^\top x) - \sigma(v^\top x) \right)^2$. For a training set $S \sim \mathcal{D}^n$, let $\mathfrak{R}_S(\mathcal{G})$ denote the empirical Rademacher complexity of the following function class

$$\mathcal{G} := \{x \mapsto w^\top x : \|w - v\|_2 \leq 1, \ \|v\|_2 = 1\}.$$

Then we have

$$\mathfrak{R}(\ell \circ \sigma \circ \mathcal{G}) = \mathbb{E}_{S \sim \mathcal{D}^n} \mathfrak{R}_S(\ell \circ \sigma \circ \mathcal{G}) \leq 2L^3 B_X^2 / \sqrt{n}.$$

With Lemmas 3.6, 3.8 and 3.9 in hand, the bound for the population risk follows in a straightforward manner.

*Proof of Theorem 3.3 for strictly increasing activations.* By Lemma 3.6, there exists some $w_t$ with $t < T$ and $\|w_t - v\|_2 \leq 1$ such that $\widehat{H}(w_t) \leq 6L^3 \gamma^{-2} \widehat{F}(v)$. By Lemmas 3.5 and Lemma 3.8, this implies that with probability at least $1 - \delta/2$,

$$\widehat{G}(w_t) \leq 6L^3 \gamma^{-3} \left( \mathsf{OPT} + 3an^{-1/2} \log^{1/2}(4/\delta) \right). \tag{3.10}$$

Since $\|w - v\|_2 \leq 1$ implies $\ell(w; x) = (1/2)(\sigma(w^\top x) - \sigma(v^\top x))^2 \leq L^2 B_X^2 / 2$, standard results from Rademacher complexity (e.g., Theorem 26.5 of [28]) imply that with probability at least $1 - \delta/2$,

$$G(w_t) \leq \widehat{G}(w_t) + \mathbb{E}_{S \sim \mathcal{D}^n} \mathfrak{R}_S(\ell \circ \sigma \circ \mathcal{G}) + 2L^2 B_X^2 \sqrt{\frac{2\log(8/\delta)}{n}},$$

where $\ell$ is the loss and $\mathcal{G}$ is the function class defined in Lemma 3.9. We can combine (3.10) with Lemma 3.9 and a union bound to get that with probability at least $1 - \delta$,

$$G(w_t) \leq 6L^3 \gamma^{-3} \left( \mathsf{OPT} + 3a\sqrt{\frac{\log(4/\delta)}{n}} \right) + \frac{2L^3 B_X^2}{\sqrt{n}} + \frac{2L^2 B_X^2 \sqrt{2\log(8/\delta)}}{\sqrt{n}}.$$

This shows that $G(w_t) \leq O(\mathsf{OPT} + n^{-1/2})$. By Claim 3.4, we have

$$F(w_t) \leq 2G(w_t) + 2\mathsf{OPT} \leq O(\mathsf{OPT} + n^{-1/2}),$$

completing the proof for those $\sigma$ satisfying Assumption 3.1. □

## 3.2 ReLU activation

The proof above crucially relies upon the fact that $\sigma$ is strictly increasing so that we may apply Fact 3.7 in the proof of Lemma 3.6. In particular, it is difficult to show a strong lower bound for the gradient direction term in (3.8) if it is possible for $(z_1 - z_2)^2$ to be arbitrarily large when $(\sigma(z_1) - \sigma(z_2))^2$ is small. To get around this, we will use the same proof technique wherein we show that the gradient lower bound involves a term that relates the auxiliary error $\widehat{H}(w_t)$ to $\widehat{F}(v)$, but our bound will involve a term of the form $O(\widehat{F}(v)^{1/2})$ rather than $O(\widehat{F}(v))$. The proof of Lemma 3.10 is in Appendix C.

**Lemma 3.10.** Suppose that $\|x\|_2 \leq B_X$ a.s. under $\mathcal{D}_x$. Suppose $\sigma$ is non-decreasing and $L$-Lipschitz. Assume $\widehat{F}(v) \in (0,1)$. For any initialization $w_0$, gradient descent with fixed step size $\eta \leq (1/4)L^{-2}B_X^{-2}$ finds weights $w_t$ satisfying $\widehat{H}(w_t) \leq 2L^2 B_X \widehat{F}(v)^{1/2}$ within $T = \lceil \eta^{-1} L^{-1} B_X^{-1} \widehat{F}(v)^{-1/2} \rceil$ iterations, with $\|w_t - v\|_2 \leq \|w_0 - v\|_2$ for each $t = 0, \dots, T-1$.

With this lemma, the proof of Theorem 3.3 for the ReLU activation follows just as in the strictly increasing case. The details are left for Appendix C.

**Remark 3.11.** An examination of the proof of Theorem 3.3 shows that when $\sigma$ satisfies Assumption 3.1, any initialization with $\|w_0 - v\|_2$ bounded by a universal constant will suffice. In particular, if we use Gaussian initialization $w_0 \sim N(0, \tau^2 I_d)$ for $\tau^2 = O(1/d)$, then by concentration of the chi-square distribution the theorem holds with (exponentially) high probability over the random initialization. For ReLU, initialization at the origin greatly simplifies the proof since Lemma 3.10 shows that $\|w_t - v\|_2 \leq \|w_0 - v\|_2$ for all $t$. When $w_0 = 0$, this implies $\|w_t - v\|_2 \leq 1$ and allows for an easy application of Lemma 3.5 to show that $H(w_t)$ is small implies $F(w_t)$ is small. For isotropic Gaussian initialization, one can show that with probability approaching 1/2 that $\|w_0 - v\|_2 < 1$ provided its variance satisfies $\tau^2 = O(1/d)$ (see e.g. Lemma 5.1 of Yehudai and Shamir [35]). In this case, the theorem will hold with constant probability over the random initialization.

## 4 Noisy teacher network setting

In this section, we consider the teacher network setting, where the joint distribution of $(x, y) \sim \mathcal{D}$ is given by a target neuron $v$ (with $\|v\|_2 \leq 1$) plus zero-mean $s$-sub-Gaussian noise,

$$y|x \sim \sigma(v^\top x) + \xi, \quad \mathbb{E}\xi|x = 0.$$

We assume throughout this section that $\xi \not\equiv 0$; we deal with the realizable setting separately (and achieve improved sample complexity) in Appendix E. We note that this is precisely the setup of the generalized linear model with (inverse) link function $\sigma$. We further note that we only assume that $\mathbb{E}[y|x] = \sigma(v^\top x)$, i.e., the noise is *not* assumed to be independent of the input $x$, and thus falls into the probabilistic concept learning model of Kearns and Schapire [22].

With the additional structural assumption of a noisy teacher, we can improve the agnostic result from $O(\mathsf{OPT}) + \varepsilon$ (for strictly increasing activations) and $O(\mathsf{OPT}^{1/2}) + \varepsilon$ (for ReLU) to $\mathsf{OPT} + \varepsilon$. The key difference from the proof in the agnostic setting is that when trying to show the gradient points in a good direction as in (3.8) and (C.2), since we know $\mathbb{E}[y|x] = \sigma(v^\top x)$, the average of terms of the form $a_i(\sigma(v^\top x_i) - y_i)$ with fixed and bounded coefficients $a_i$ will concentrate around zero. This allows us to improve the lower bound from $\langle \nabla \widehat{F}(w_t), w_t - v \rangle \geq C(\widehat{H}(w) - \widehat{F}(v)^\alpha)$ to one of the form $\geq C(\widehat{H}(w) - \varepsilon)$, where $C$ is an absolute constant. The full proof of Theorem 4.1 is given in Appendix D.

**Theorem 4.1.** Suppose $\mathcal{D}_x$ satisfies $\|x\|_2 \leq B_X$ a.s. and $\mathbb{E}[y|x] = \sigma(v^\top x)$ for some $\|v\|_2 \leq 1$. Assume that $\sigma(v^\top x) - y$ is $s$-sub-Gaussian. Assume gradient descent is initialized at $w_0 = 0$ and fix a step size $\eta \leq (1/4)L^{-2}B_X^{-2}$. If $\sigma$ satisfies Assumption 3.1, let $\gamma$ be the constant corresponding to $\rho = 2B_X$. There exists an absolute constant $c_0 > 0$ such that for any $\delta > 0$, with probability at least

$1 - \delta$, gradient descent for $T = \eta^{-1}\sqrt{n}/(c_0 L B_x s \sqrt{\log(4d/\delta)})$ iterations finds weights $w_t$, $t < T$, satisfying

$$F(w_t) \leq \mathsf{OPT} + C_1 n^{-1/2} + C_2 n^{-1/2}\sqrt{\log(8/\delta)} + C_3 n^{-1/2}\sqrt{\log(4d/\delta)}, \qquad (4.1)$$

where $C_1 = 4L^3 B_X^2$, $C_2 = 2\sqrt{2}L^2 B_X^2 \sqrt{2}$, and $C_3 = 4c_0\gamma^{-1}L^2 s B_X$. When $\sigma$ is ReLU, further assume that $\mathcal{D}_x$ satisfies Assumption 3.2 for constants $\alpha, \beta > 0$, and let $\nu = \alpha^4\beta/8\sqrt{2}$. Then (4.1) holds for $C_1 = B_X^2\nu^{-1}$, $C_2 = 2C_1$, and $C_3 = 4c_0 s\nu^{-1}B_X$.

We first note that although (4.1) contains a $\log(d)$ term, the dependence on the dimension can be removed if we assume that the noise is bounded rather than sub-Gaussian; details for this are given in Appendix D. As mentioned previously, if we are in the realizable setting, i.e. $\xi \equiv 0$, we can improve the sample and runtime complexities to $O(\varepsilon^{-1})$ by using online SGD and a martingale Bernstein bound. For details on the realizable case, see Appendix E.

In comparison with existing literature, Kakade et al. [20] proposed GLMTron to show the learnability of the noisy teacher network for a non-decreasing and Lipschitz activation $\sigma$ when the noise is bounded.[2] In GLMTron, updates take the form $w_{t+1} = w_t - \eta\tilde{g}_t$ where $\tilde{g}_t = (\sigma(w_t^\top x) - y)x$, while in gradient descent, the updates take the form $w_{t+1} = w_t - \eta g_t$ where $g_t = \tilde{g}_t\sigma'(w_t^\top x)$. Intuitively, when the weights are in a bounded region and $\sigma$ is strictly increasing and Lipschitz, the derivative satisfies $\sigma'(w_t^\top x) \in [\gamma, L]$ and so the additional $\sigma'$ factor will not significantly affect the algorithm. For ReLU this is more complicated as the gradient could in the worst case be zero in a large region of the input space, preventing effective learnability using gradient-based optimization, as was demonstrated in the negative result of Yehudai and Shamir [35]. For this reason, a type of nondegeneracy condition like Assumption 3.2 is essential for gradient descent on ReLUs.

Our GLM result is also comparable to recent work by Foster et al. [11], where the authors provide a meta-algorithm for translating guarantees for $\varepsilon$-stationary points of the empirical risk to guarantees for the population risk provided that the population risk satisfies the so-called "gradient domination" condition and the algorithm can guarantee that the weights remain bounded (see their Proposition 3). By considering GLMs with bounded, strictly increasing, Lipschitz activations, they show the gradient domination condition holds, and any algorithm that can find a stationary point of an $\ell^2$-regularized empirical risk objective is guaranteed to have a population risk bound. In contrast, our result concretely shows that vanilla gradient descent learns the GLM, even in the ReLU setting.

## 5 Conclusion and remaining open problems

In this work, we considered the problem of learning a single neuron with the squared loss by using gradient descent on the empirical risk. We first analyzed this in the agnostic PAC learning framework and showed that if the activation function is strictly increasing and Lipschitz, then gradient descent finds weights with population risk $O(\mathsf{OPT}) + \varepsilon$, where $\mathsf{OPT}$ is the smallest possible population risk achieved by a single neuron. When the activation function is ReLU, we showed that gradient descent finds a point with population risk at most $O(\mathsf{OPT}^{1/2}) + \varepsilon$. Under the more restricted noisy teacher network setting, we showed the population risk guarantees improve to $\mathsf{OPT} + \varepsilon$ for both strictly increasing activations and ReLU.

Our work points towards a number of open problems. Does gradient descent on the empirical risk provably achieve population risk with a better dependence on $\mathsf{OPT}$ than we have shown in this work, or are there distributions for which this is impossible? Recent work by Goel et al. [15] provides a statistical query lower bound for learning a sigmoid with respect to the correlation loss $\mathbb{E}[\ell(y\sigma(w^\top x))]$, but we are not aware of lower bounds for learning non-ReLU single neurons under the squared loss. It thus remains a possibility that gradient descent (or another algorithm) can achieve $\mathsf{OPT} + \varepsilon$ risk for such activation functions. For ReLU, Diakonikolas et al. [8] showed that gradient descent on a convex surrogate for the empirical risk can achieve $O(\mathsf{OPT}) + \varepsilon$ population risk for log concave distributions; it would be interesting if such bounds could be shown for gradient descent on the empirical risk itself.

## Broader Impact

This paper provides a theoretical analysis of gradient descent when used for learning a single neuron. As a theoretical work, its potential risk for negative societal impacts is extremely limited. On the other hand, our general lack of understanding of why gradient descent on large neural networks can find weights that have both small empirical risk and also small population risk is worrisome given the widespread adoption of large neural networks in sensitive technology applications. (A reasonable expectation for using a piece of technology is that we understand how and why it works.) Our work helps explain how, in a simple neural network model, gradient descent can learn solutions that generalize well even though the optimization problem is highly nonconvex and nonsmooth. As such, it provides a building block for understanding how more complex neural network models can be learned by gradient descent.

## Acknowledgement

We thank Adam Klivans for his helpful comments on our work. We also thank Surbhi Goel for pointing out how to extend the results of Diakonikolas et al. [8] to more general distributions and to leaky-ReLU-type activation functions. This research was sponsored in part by the National Science Foundation CAREER Award 1906169, IIS-2008981 and Salesforce Deep Learning Research Award. The views and conclusions contained in this paper are those of the authors and should not be interpreted as representing any funding agencies.

## Footnotes

[1]We note that for some distributions, the $B_X$ term may hide an implicit dependence on $d$; more detailed comments on this are given in Appendix A.

[2]A close inspection of the proof shows that sub-Gaussian noise can be handled with the same concentration of norm sub-Gaussian random vectors that we use for our results.

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
