[Supplementary Material]

## A  Detailed comparisons with related work

Here, we describe comparisons of our results to those in the literature and give detailed comments on the specific rates we achieve. In Table 1, we compare our agnostic learning results. We note the guarantees for the population risk in the fourth column, the marginal distributions over $x$ for which the bounds hold in the fifth column, and the sample complexity required to reach the specified level of risk plus some $\varepsilon > 0$ in the final column. Our results in this setting come from Theorem 3.3. The Big-O notation hides constants that may depend on the parameters of the distribution or activation function, but does not hide explicit dependence on the dimension $d$. However, the parameters of the distribution itself may have *implicit* dependence on the dimension. In particular, for bounded distributions that satisfy $\|x\|_2 \leq B_X$, the $O()$ hides multiplicative factors that depend on $B_X$. This means that if $B_X$ depends on $d$, so will our bounds. For ReLU, the $O()$ hides polynomial factors in $B_X$. For non-ReLU, the worst-case activation functions under consideration in Assumption 3.1 (e.g. the sigmoid) can have $\gamma \sim \exp(-B_X)$, making the runtime and sample complexity depend on $\gamma^{-1} \sim \exp(B_X)$, in which case it is preferable for $B_X$ to be a constant independent of the dimension. We note that the sample complexity for Diakonikolas et al. [8] for the $(1 + \delta)\mathsf{OPT}$ guarantee is $O(\varepsilon^{-2}[d\delta^{-3}\nu^{-2}]^{\delta^{-3}})$ when $\mathcal{D}_x$ is $\nu$ sub-Gaussian for some $\nu = O(1)$, and thus the exact dependence on the dimension depends on the sub-Gaussian norm and error threshold desired.

In Table 2, we provide comparisons of our noisy teacher network setting, where $y = \sigma(v^\top x) + \xi$ for some zero mean noise $\xi$. Our results in this setting come from Theorem 4.1. The complexity column here denotes the sample complexity required to reach population risk $\mathsf{OPT} + \varepsilon$. The subspace eigenvalue assumption given by Mukherjee and Muthukumar [26] is that $\mathbb{E}[xx^\top \mathbb{1}(v^\top x \geq 0)] \succ 0$; a similar assumption was used by Du et al. [10] in the realizable setting. We note that the result of Mukherjee and Muthukumar holds for any bounded noise distribution and thus is in the more general adversarial noise (but not agnostic[3]) setting.

Finally, in Table 3, we provide comparisons with results in the realizable setting ($\xi \equiv 0$). (Our results in this setting are given in Theorem E.1 in Appendix E.) For G.D. and S.G.D., the complexity column denotes the sample complexity required to reach population risk $\varepsilon$. For G.D. or gradient flow on the population risk, it refers to the runtime complexity only as there are no samples in this setting. For Du et al. [10], the subspace eigenvalue assumption is that for any $w$ and for the target neuron $v$, it holds that $\mathbb{E}[xx^\top \mathbb{1}(w^\top x \geq 0, v^\top x \geq)] \succ 0$. This is a nondegeneracy assumption that is related to the marginal spread condition given in Assumption 3.2, in the sense that it allows for one to show that $H$ is an upper bound for $G$. Finally, we note that any result in the agnostic or noisy teacher network settings applies in the realizable setting as well.

## B  Proof of Lemma 3.5

To prove Lemma 3.5, we use the following result of Yehudai and Shamir [35].

**Lemma B.1** (Lemma B.1, [35])**.** Under Assumption 3.2, for any two vectors $a, b \in \mathbb{R}^2$ satisfying $\theta(a, b) \leq \pi - \delta$ for $\delta \in (0, \pi]$, it holds that

$$\inf_{u \in \mathbb{R}^2: \|u\|=1} \int (u^\top y)^2 \mathbb{1}(a^\top y \geq 0, \ b^\top y \geq 0, \ \|y\| \leq \alpha) dy \geq \frac{\alpha^4}{8\sqrt{2}} \sin^3(\delta/4).$$

*Proof of Lemma 3.5.* We first consider the case when $\sigma$ satisfies Assumption 3.1. By assumption,

$$H(w) = (1/2)\mathbb{E}\left[\left(\sigma(w_t^\top x) - \sigma(v^\top x)\right)^2 \sigma'(w_t^\top x)\right] \leq \varepsilon.$$

Table 1: Comparison of results in the agnostic setting

| Algorithm | Activations | Pop. risk | $\mathcal{D}_x$ | Sample Complexity |
|---|---|---|---|---|
| Halfspace reduction [14] | ReLU | $O(\mathsf{OPT}^{2/3})$ | standard Gaussian | $O(\mathrm{poly}(d, \varepsilon^{-1}))$ |
| Convex surrogate G.D. [8][4] | ReLU | $O(\mathsf{OPT})$ | isotropic +log-concave | $O(d\varepsilon^{-2})$ |
| Convex surrogate G.D. + Domain Partition [8] | ReLU | $(1 + \delta)\mathsf{OPT}$ | sub-Gaussian | $O(d^c\varepsilon^{-2})$ |
| Gradient Descent (This paper) | strictly increasing + Lipschitz | $O(\mathsf{OPT})$ | bounded | $O(\varepsilon^{-2})$ |
| Gradient Descent (This paper) | ReLU | $O(\mathsf{OPT}^{1/2})$ | bounded + marginal spread | $O(\varepsilon^{-4})$ |

Table 2: Comparison of results in the noisy teacher network setting

| Paper | Algorithm | Activations | $\mathcal{D}_x$ | Sample Complexity |
|---|---|---|---|---|
| K. et al. [20] | GLMTron | increasing + Lipschitz | bounded | $O(\varepsilon^{-2})$ |
| M. & M. [26] | Modified S.G.D. | ReLU | bounded + subspace eigenvalue | $O(\log(1/\varepsilon))$ |
| F. et al. [11] | Meta-algo. | strictly increasing + Lipschitz + $\sigma'$ Lipschitz | bounded | $O(\varepsilon^{-2} \wedge d\varepsilon^{-1})$ |
| M. et al. [25] | G.D. | strictly increasing + diff'ble + Lipschitz + $\sigma'$ Lipschitz + $\sigma''$ Lipschitz | centered + sub-Gaussian + $\mathbb{E}[xx^\top] \succ 0$ | $O(d\varepsilon^{-1})$ |
| This paper | G.D. | strictly increasing + Lipschitz | bounded | $O(\varepsilon^{-2})$ |
| This paper | G.D. | ReLU | bounded + marginal spread | $O(\varepsilon^{-2})$ |

Since the term in the expectation is nonnegative, restricting the integral to a smaller set only decreases its value, so that

$$(1/2)\mathbb{E}\left[\left(\sigma(w_t^\top x) - \sigma(v^\top x)\right)^2 \sigma'(w_t^\top x)\mathbb{1}(|w_t^\top x| \le \rho)\right] \le \varepsilon. \tag{B.1}$$

For $\rho = BW$, since $\|w\|_2 \le W$, the inclusion $\{\|x\|_2 \le \rho/W\} = \{\|x\|_2 \le B\} \subset \{|w_t^\top x| \le \rho\}$ holds. This means we can lower bound (B.1) by substituting the indicator $\mathbb{1}(|w_t^\top x| \le \rho)$ with $\mathbb{1}(\|x\|_2 \le B)$, which is identically one by assumption. Since $H(w) \le \varepsilon$, this implies

$$\frac{\gamma}{2}\mathbb{E}\left[\left(\sigma(w_t^\top x) - \sigma(v^\top x)\right)^2\right] \le (1/2)\mathbb{E}\left[\left(\sigma(w_t^\top x) - \sigma(v^\top x)\right)^2 \sigma'(w_t^\top x)\mathbb{1}(\|x\|_2 \le B)\right] \le \varepsilon.$$

Table 3: Comparison of results in the realizable setting

| Paper | Algorithm | Activations | $\mathcal{D}_x$ | Sample Complexity |
|---|---|---|---|---|
| D. et al. [10] | S.G.D. | ReLU | bounded + subspace eigenvalue | $O(\log(1/\varepsilon))$ |
| S. [30] | Projected Regularized G.D. | ReLU | standard Gaussian | $O(\log(1/\varepsilon))$ |
| Y. & S. [35] | Pop. G.D. | leaky ReLU | bounded + $\mathbb{E}[xx^\top] \succ 0$ | $O(\log(1/\varepsilon))$ |
| Y. & S. | Pop. G.D. | $\inf_{0<z<\alpha} \sigma'(z) > 0$ + Lipschitz | bounded + marginal spread | $O(\log(1/\varepsilon))$ |
| Y. & S. | Population Gradient Flow | ReLU | marginal spread + spherical symmetry | $O(\log(1/\varepsilon))$ |
| Y. & S. | S.G.D. | $\inf_{0<z<\alpha} \sigma'(z) > 0$ + Lipschitz | bounded + marginal spread | $\tilde{O}(\varepsilon^{-2})$ |
| This paper | Pop. G.D. + S.G.D. | strictly increasing + Lipschitz | bounded | $O(\varepsilon^{-1})$ |
| This paper | Pop. G.D. + S.G.D. | ReLU | bounded + marginal spread | $O(\varepsilon^{-1})$ |

Dividing both sides by $\gamma$ completes this part of the proof.

For ReLU, let us assume that $H(w) \leq \varepsilon$, and denote the event

$$K_{w,v} := \{w^\top x \geq 0, v^\top x \geq 0\},$$

and define $\zeta := \beta\alpha^4/8\sqrt{2}$. Since $H(w) = \mathbb{E}[(\sigma(w^\top x) - \sigma(v^\top x))^2 \mathbb{1}(w^\top x \geq 0)] \leq \zeta\varepsilon$, it holds that

$$\mathbb{E}\left[\left(\sigma(w^\top x) - \sigma(v^\top x)\right)^2 \mathbb{1}(K_{w,v})\right] \leq \zeta\varepsilon. \tag{B.2}$$

Denote $\widehat{w}$ and $\widehat{v}$ as the projections of $w$ and $v$ respectively onto the two dimensional subspace $\mathrm{span}(w,v)$. Using a proof similar to that of Yehudai and Shamir [35], we have

$$\mathbb{E}_{x\sim\mathcal{D}}\left[\left(w^\top x - v^\top x\right)^2 \mathbb{1}(K_{w,v})\right] = \|w-v\|_2^2 \, \mathbb{E}_{x\sim\mathcal{D}}\left[\left(\left(\frac{w-v}{\|w-v\|_2}\right)^\top x\right)^2 \mathbb{1}(K_{w,v})\right]$$

$$\geq \|w-v\|_2^2 \inf_{u\in\mathrm{span}(w,v),\, \|u\|=1} \mathbb{E}_x\left[\mathbb{1}(u^\top x)^2 \mathbb{1}(K_{w,v})\right]$$

$$= \|w-v\|_2^2 \inf_{u\in\mathbb{R}^2,\, \|u\|=1} \mathbb{E}_{y\sim\mathcal{D}_{w,v}}\left[(u^\top y)^2 \mathbb{1}(\widehat{w}^\top y \geq 0,\, \widehat{v}^\top y \geq 0)\right]$$

$$\geq \|w-v\|_2^2 \inf_{u\in\mathbb{R}^2,\, \|u\|=1} \int (u^\top y)^2 \mathbb{1}(\widehat{w}^\top y \geq 0,\, \widehat{v}^\top y \geq 0,\, \|y\|_2 \leq \alpha)p_{w,v}(y)dy$$

$$\geq \beta \|w-v\|_2^2 \inf_{u\in\mathbb{R}^2,\, \|u\|=1} \int (u^\top y)^2 \mathbb{1}(\widehat{w}^\top y \geq 0,\, \widehat{v}^\top y \geq 0,\, \|y\|_2 \leq \alpha)dy. \tag{B.3}$$

By assumption, $\|w-v\|_2 \leq 1$. Since

$$1 \geq \|w-v\|_2^2 = \|w\|_2 \left(\|w\|_2 - 2\cos\theta(w,v)\right) + 1,$$

we must have either $w = 0$ or $\theta(w,v) \in [0, \pi/2]$. To see that $w = 0$ is impossible, suppose for the contradiction that $w = 0$ and so $H(w) = H(0) \leq \zeta\varepsilon$. Let $z$ be any vector orthogonal to $v$, so that

$\theta(v, z) = \pi/2$. Then,

$$
\begin{aligned}
\zeta\varepsilon &\geq H(0) \\
&= \mathbb{E}_{x\sim\mathcal{D}}\left[(v^\top x)^2 \mathbb{1}(v^\top x \geq 0)\right] \\
&= \mathbb{E}_{y\sim\mathcal{D}_{0,v}}\left[(\widehat{v}^\top y)^2 \mathbb{1}(\widehat{v}^\top y \geq 0)\right] \\
&\geq \inf_{u:\,\|u\|=1} \int (u^\top x)^2 \mathbb{1}(v^\top x \geq 0, z^\top x \geq 0, \|y\|_2 \leq \alpha) p_{0,v}(y)dy \\
&\geq \beta \inf_{u:\,\|u\|=1} \int (u^\top x)^2 \mathbb{1}(v^\top x \geq 0, z^\top x \geq 0, \|y\|_2 \leq \alpha)dy \\
&\geq \frac{\beta\alpha^4}{8\sqrt{2}}.
\end{aligned}
\tag{B.4}
$$

The last line follows by using Lemma B.1. For $\varepsilon < 1$, this is impossible by the definition of $\zeta$. This shows that $\theta(w, v) \leq \pi/2$. We can therefore apply Lemma B.1 to (B.3) to get

$$
\begin{aligned}
\zeta\varepsilon &\geq \beta\|w - v\|_2^2 \inf_{u\in\mathbb{R}^2,\,\|u\|=1} \int (u^\top y)^2 \mathbb{1}(\widehat{w}^\top y \geq 0,\ \widehat{v}^\top y \geq 0,\ \|y\|_2 \leq \alpha)dy \\
&\geq \frac{\beta\alpha^4}{8\sqrt{2}}\|w - v\|_2^2 \\
&= \zeta B^2 \|w - v\|_2^2.
\end{aligned}
$$

This shows that $\|w - v\|_2^2 \leq B^{-2}\varepsilon$. Since $\sigma$ is 1-Lipschitz, Hölder's inequality and $\mathbb{E}\|x\|_2^2 \leq B^2$ imply that $G(w) \leq \varepsilon$. $\qquad\square$

## C  Proofs for ReLU activation in agnostic setting

In this section we prove Lemma 3.10, which we reproduce below for the reader's convenience.

**Lemma C.1.** Suppose that $\|x\|_2 \leq B_X$ a.s. under $\mathcal{D}_x$. Suppose $\sigma$ is non-decreasing and $L$-Lipschitz. Assume $\widehat{F}(v) \in (0, 1)$. Gradient descent with fixed step size $\eta \leq (1/4)L^{-2}B_X^{-2}$ initialized at $w_0 = 0$ finds weights $w_t$ satisfying $\widehat{H}(w_t) \leq 2L^2 B_X \widehat{F}(v)^{1/2}$ within $T = \lceil \eta^{-1}L^{-1}B_X^{-1}\widehat{F}(v)^{-1/2} \rceil$ iterations, with $\|w_t - v\|_2 \leq 1$ for each $t = 0, \dots, T-1$.

As mentioned in the main section, the proof of Lemma 3.6 heavily relied upon Fact 3.7, which is only satisfied if the activation is *strictly* increasing. When $\sigma$ is only *non-decreasing*, we can get a similar lower bound as we did in (3.8) if we use the following fact.

**Fact C.2.** If $\sigma$ is non-decreasing and $L$-Lipschitz, then for any $z_1, z_2$ in the domain of $\sigma$, it holds that $(\sigma(z_1) - \sigma(z_2))(z_1 - z_2) \geq L^{-1}(\sigma(z_1) - \sigma(z_2))^2$.

With this fact we can present the analogue to Lemma 3.6 that holds for a general non-decreasing and Lipschitz activation and hence includes the ReLU.

*Proof.* Just as in the proof of Lemma 3.6, the lemma is proved if we can show that for every $t \in \mathbb{N}$, either (a) $\widehat{H}(w_\tau) \leq 2L^2 B_X \widehat{F}(v)^{1/2}$ for some $\tau < t$, or (b) $\|w_t - v\|_2^2 \leq \|w_{t-1} - v\|_2^2 - \eta L B_X \widehat{F}(v)^{1/2}$ holds. To this end we assume the induction hypothesis holds for some $t \in \mathbb{N}$, and since we are done if (a) holds, we assume (a) does not hold and thus for every $\tau \leq t$, we have $\widehat{H}(w_\tau) > 2L^2 B_X \widehat{F}(v)^{1/2}$. Since (a) does not hold, $\|w_\tau - v\|_2^2 \leq \|w_{\tau-1} - v\|_2^2 - \eta L B_X \widehat{F}(v)^{1/2}$ holds for each $\tau = 1, \dots, t$ and hence the identity

$$
\|w_\tau - v\|_2 \leq 1 \quad \forall \tau \leq t,
\tag{C.1}
$$

holds. We now proceed with showing the analogues of (3.8) and (3.9). We begin with the lower bound,

$$
\left\langle \nabla\widehat{F}(w_t), w_t - v \right\rangle = (1/n)\sum_{i=1}^{n}\left(\sigma(w_t^\top x_i) - \sigma(v^\top x_i)\right)\sigma'(w_t^\top x_i)(w_t^\top x_i - v^\top x_i)
$$

$$+ \left\langle (1/n) \sum_{i=1}^{n} \left( \sigma(v^\top x_i) - y_i \right) \sigma'(w_t^\top x_i) x_i, w_t - v \right\rangle \tag{C.2}$$

$$\geq (1/Ln) \sum_{i=1}^{n} \left( \sigma(w_t^\top x_i) - \sigma(v^\top x_i) \right)^2 \sigma'(w_t^\top x_i)$$

$$- \|w_t - v\|_2 \left\| (1/n) \sum_{i=1}^{n} \left( \sigma(v^\top x_i) - y_i \right) \sigma'(w_t^\top x_i) x_i \right\|_2$$

$$\geq 2L^{-1} \widehat{H}(w_t) - L B_X \widehat{F}(v)^{1/2}. \tag{C.3}$$

In the first inequality, we have used Fact C.2 and that $\sigma'(z) \geq 0$ for the first term. For the second term, we use Cauchy–Schwarz. The last inequality is a consequence of (C.1), Cauchy–Schwarz, and that $\sigma'(z) \leq L$ and $\|x\|_2 \leq B_X$. As for the gradient upper bound at $w_t$, the bound (3.9) still holds since it only uses that $\sigma$ is $L$-Lipschitz. The choice of $\eta \leq (1/4) L^{-2} B_X^{-2}$ then ensures

$$\begin{aligned}
\|w_t - v\|_2^2 - \|w_{t+1} - v\|_2^2 &\geq 2\eta \left( 2L^{-1} \widehat{H}(w_t) - L B_X \widehat{F}(v)^{1/2} \right) \\
&\quad - \eta^2 \left( 4 B_X^2 L \widehat{H}(w_t) + 4 L^2 B_X^2 \widehat{F}(v) \right) \\
&\geq \eta \left( 3 L^{-1} \widehat{H}(w_t) - 3 L B_X \left( \widehat{F}(v) \vee \widehat{F}(v)^{1/2} \right) \right) \\
&\geq \eta L B_X \widehat{F}(v)^{1/2}, \tag{C.4}
\end{aligned}$$

where the last line comes from the induction hypothesis that $\widehat{H}(w_t) \geq 2 L^2 B_X \widehat{F}(v)^{1/2}$ and since $\widehat{F}(v) \in (0, 1)$. This completes the proof. $\qquad\square$

With the above lemma, we can prove the ReLU case of Theorem 3.3.

*Proof of Theorem 3.3 for ReLU.* We highlight here the main technical differences with the proof for the strictly increasing case. Although Lemma 3.9 applies to the loss function $\ell(w; x) = (1/2) \left( \sigma(w^\top x) - \sigma(v^\top x) \right)^2$, the same results hold for the loss function $\tilde{\ell}(w; x) = \ell(w; x) \sigma'(w^\top x)$ for ReLU, since $\nabla \sigma'(w^\top x) \equiv 0$ a.e. Thus $\tilde{\ell}$ is still $B_X$-Lipschitz, and we have

$$\mathbb{E}_{S \sim \mathcal{D}^n} \mathfrak{R}_S \left( \tilde{\ell} \circ \sigma \circ \mathcal{G} \right) \leq \frac{2 B_X^2}{\sqrt{n}}. \tag{C.5}$$

With this in hand, the proof is essentially identical: By Lemmas 3.10 and 3.8, with probability at least $1 - \delta/2$ gradient descent finds a point with

$$\widehat{H}(w_t) \leq 2 B_X \widehat{F}(v)^{1/2} \leq 2 B_X \left( \mathsf{OPT}^{1/2} + \frac{\sqrt{3a} \log^{1/4}(4/\delta)}{n^{1/4}} \right). \tag{C.6}$$

We can then use (C.5) to get that with probability at least $1 - \delta$,

$$H(w_t) \leq 2 B_X \left( \mathsf{OPT}^{1/2} + \frac{\sqrt{3a} \log^{1/4}(4/\delta)}{n^{1/4}} \right) + \frac{2 B_X^2}{\sqrt{n}} + 2 B_X^2 \sqrt{\frac{2 \log(8/\delta)}{n}}. \tag{C.7}$$

Since $\mathcal{D}_x$ satisfies Assumption 3.2 and $\|w_t - v\|_2 \leq 1$, Lemma 3.5 yields $G(w_t) \leq 8 \sqrt{2} \alpha^{-4} \beta^{-1} H(w_t)$. Then applying Claim 3.4 completes the proof. $\qquad\square$

# D  Noisy teacher network proofs

As in the agnostic case, we have a key lemma that shows $\widehat{H}$ is small when we run gradient descent for a sufficiently large time. Note that one difference with the proof in the agnostic case is that we do not need to consider different auxiliary errors for the strictly increasing and ReLU cases; $H$ alone suffices.

**Lemma D.1.** Suppose that $\|x\|_2 \leq B_X$ a.s. under $\mathcal{D}_x$. Let $\sigma$ be non-decreasing and $L$-Lipschitz. Suppose that the bound

$$\|(1/n) \sum_{i=1}^{n} \left(\sigma(v^\top x_i) - y_i\right) \alpha_i x_i\|_2 \leq K \leq 1. \tag{D.1}$$

holds for scalars satisfying $\alpha_i \in [0, L]$. Then gradient descent run with fixed step size $\eta \leq (1/4)L^{-2}B_X^{-2}$ from initialization $w_0 = 0$ finds weights $w_t$ satisfying $\widehat{H}(w_t) \leq 4LK$ within $T = \lceil \eta^{-1}K^{-1} \rceil$ iterations, with $\|w_t - v\|_2 \leq 1$ for each $t = 0, \ldots, T-1$.

*Proof.* Just as in the proof of Lemma 3.6, the theorem can be shown by proving the following induction statement. We claim that for every $t \in \mathbb{N}$, either (a) $\widehat{H}(w_\tau) \leq 4LK$ for some $\tau < t$, or (b) $\|w_t - v\|_2^2 \leq \|w_{t-1} - v\|_2^2 - \eta K$. If the induction hypothesis holds, then until gradient descent finds a point where $\widehat{H}(w_t) \leq 4LK$, the squared distance $\|w_t - v\|_2^2$ decreases by $\eta K$ at every iteration. Since $\|w_0 - v\|_2^2 = 1$, this means there can be at most $\eta^{-1}K^{-1}$ iterations until we reach $\widehat{H}(w_t) \leq 4LK$. This shows the induction statement implies the theorem.

We begin with the proof by supposing the induction hypothesis holds for $t$, and considering the case $t + 1$. If (a) holds, then we are done. So now consider the case that for every $\tau \leq t$, we have $\widehat{H}(w_\tau) > 4LK$. Since (a) does not hold, $\|w_\tau - v\|_2^2 \leq \|w_{\tau-1} - v\|_2^2 - \eta K$ holds for each $\tau = 1, \ldots, t$. Since $\|w_0 - v\|_2 = 1$, this implies

$$\|w_\tau - v\|_2 \leq 1 \; \forall \tau \leq t. \tag{D.2}$$

We can therefore bound

$$
\begin{aligned}
\left\langle \nabla \widehat{F}(w_t), w_t - v \right\rangle &= \left\langle \frac{1}{n} \sum_{1=1}^{n} \left(\sigma(w_t^\top x_i) - y_i\right) \sigma'(w_t^\top x_i) x_i, w_t - v \right\rangle \\
&= \frac{1}{n} \sum_{i=1}^{n} \left(\sigma(w_t^\top x_i) - \sigma(v^\top x_i)\right) \sigma'(w_t^\top x_i)(w_t^\top x_i - v^\top x_i) \\
&\quad + \left\langle \frac{1}{n} \sum_{i=1}^{n} \left(\sigma(v^\top x_i) - y_i\right) \sigma'(w_t^\top x_i) x_i, w_t - v \right\rangle \\
&\geq \frac{L^{-1}}{n} \sum_{i=1}^{n} \left(\sigma(w_t^\top x_i) - \sigma(v^\top x_i)\right)^2 \sigma'(w_t^\top x_i) - K \|w_t - v\|_2 \\
&\geq 2L^{-1}\widehat{H}(w_t) - K. \tag{D.3}
\end{aligned}
$$

In the first inequality, we have used Fact C.2 for the first term. For the second term, we use (D.1) and that $\alpha_i := \sigma'(w_t^\top x_i) \in [0, L]$. The last inequality uses (D.2).

For the gradient upper bound, we have

$$
\begin{aligned}
\left\|\nabla \widehat{F}(w_t)\right\|_2^2 &= \left\| \frac{1}{n} \sum_{i=1}^{n} \left(\sigma(w_t^\top x_i) - \sigma(v^\top x_i)\right) \sigma'(w_t^\top x_i) x_i + \frac{1}{n} \sum_{i=1}^{n} \left(\sigma(v^\top x_i) - y_i\right) \sigma'(w_t^\top x_i) x_i \right\|_2^2 \\
&\leq 2 \left\| \frac{1}{n} \sum_{i=1}^{n} \left(\sigma(w_t^\top x_i) - \sigma(v^\top x_i)\right) \sigma'(w_t^\top x_i) x_i \right\|_2^2 \\
&\quad + 2 \left\| \frac{1}{n} \sum_{i=1}^{n} \left(\sigma(v^\top x_i) - y_i\right) \sigma'(w_t^\top x_i) x_i \right\|_2^2 \\
&\leq \frac{2LB_X^2}{n} \sum_{i=1}^{n} \left(\sigma(w^\top x_i) - \sigma(v^\top x_i)\right)^2 \sigma'(w_t^\top x_i) + 2K^2 \\
&= 4LB_X^2 \widehat{H}(w_t) + 2K^2. \tag{D.4}
\end{aligned}
$$

The first inequality uses Young's inequality. The second uses that $\sigma'(z) \leq L$ and that $\|x\|_2 \leq B_X$ a.s. and (D.1).

Putting (D.3) and (D.4) together, the choice of $\eta \leq (1/4)L^{-2}B_X^{-2}$ gives us

$$\|w_t - v\|_2^2 - \|w_{t+1} - v\|_2^2 = 2\eta \left\langle \nabla \widehat{F}(w_t), w_t - v \right\rangle - \eta^2 \left\| \nabla \widehat{F}(w_t) \right\|_2^2$$
$$\geq 2\eta(L^{-1}\widehat{H}(w_t) - K) - \eta^2 \left( 4LB_X^2 \widehat{H}(w_t) + 2K^2 \right)$$
$$\geq \eta L^{-1}\widehat{H}(w_t) - 3\eta K.$$

In particular, this implies

$$\|w_{t+1} - v\|_2^2 \leq \|w_t - v\|_2^2 + 3\eta K - \eta L^{-1}\widehat{H}(w_t) \tag{D.5}$$

Since $\widehat{H}(w_t) > 4KL$, this completes the induction. The base case follows easily since $\|w_0 - v\|_2 = 1$ allows for us to deduce the desired bound on $\|w_1 - v\|_2^2$ using (D.5). $\qquad\square$

To prove a concrete bound on the $K$ term of Lemma D.1, we will need the following definition of norm sub-Gaussian random vectors.

**Definition D.2.** A random vector $z \in \mathbb{R}^d$ is said to be *norm sub-Gaussian with parameter $s > 0$* if

$$\mathbb{P}(\|z - \mathbb{E}z\| \geq t) \leq 2\exp(-t^2/2s^2).$$

A Hoeffding-type inequality for norm sub-Gaussian vectors was recently shown by Jin et al. [19].

**Lemma D.3** (Lemma 6, [19]). Suppose $z_1, \ldots, z_n \in \mathbb{R}^d$ are random vectors with filtration $\mathcal{F}_t := \sigma(z_1, \ldots, z_t)$ such that $z_i | \mathcal{F}_{i-1}$ is a zero-mean norm sub-Gaussian vector with parameter $s_i \in \mathbb{R}$ for each $i$. Then, there exists an absolute constant $c > 0$ such that for any $\delta > 0$, with probability at least $1 - \delta$,

$$\left\| \sum_{i=1}^n z_i \right\| \leq c\sqrt{\log(2d/\delta) \sum_{i=1}^n s_i^2}.$$

Using this, we can show that if $\xi_i := \sigma(v^\top x_i) - y_i$ is $s$ sub-Gaussian, then we can get a bound on $K$ at rate $n^{-1/2}$. We note that if we make the stronger assumption that $\xi_i$ is bounded a.s., we can get rid of the $\log(d)$ dependence by using concentration of bounded random variables in a Hilbert space (e.g. Pinelis and Sakhanenko [27], Corollary 2).

**Lemma D.4.** Suppose that $\|x\|_2 \leq B_X$ a.s. under $\mathcal{D}_x$, and let $\sigma$ be any continuous function. Assume $\xi_i := \sigma(v^\top x_i) - y_i$ is $s$ sub-Gaussian and satisfies $\mathbb{E}[\xi_i | x_i] = 0$. Then there exists an absolute constant $c_0 > 0$ such that for constants $\alpha_i \in [0, L]$, with probability at least $1 - \delta$, we have

$$\|(1/n) \sum_{i=1}^n \left( \sigma(v^\top x_i) - y_i \right) \alpha_i x_i\| \leq c_0 L B_X s \sqrt{n^{-1}\log(2d/\delta)}.$$

*Proof of Lemma D.4.* Define $z_i := \left( \sigma(v^\top x_i) - y_i \right) \alpha_i x_i$. Using iterated expectations, we see that $\mathbb{E}[z_i] = 0$. Since $\sigma(v^\top x_i) - y_i$ is $s$-sub-Gaussian and $\|\alpha_i x_i\|_2 \leq LB_X$, it follows from the definition that $z_i$ is norm sub-Gaussian with parameter $LB_X s$ for each $i$. By Lemma D.3, we have with probability at least $1 - \delta$,

$$\left\| \sum_{i=1}^n z_i \right\| \leq c\sqrt{\log(2d/\delta)L^2 B_X^2 n s^2}.$$

Dividing each side by $n$ proves the lemma. $\qquad\square$

*Proof of Theorem 4.1.* By Lemmas D.1 and D.4, there exists some $w_t$, $t < T$ and $\|w_t - v\|_2 \leq 1$, such that

$$\widehat{H}(w_t) \leq 4LK \leq 4c_0 L^2 B_X s \sqrt{\frac{\log(2d/\delta)}{n}}.$$

Consider $\sigma$ satisfying Assumption 3.1 first, with $\gamma$ corresponding to $\rho = 2B_X$. Since $\|w_t\|_2 \leq 2$, we can use Lemma 3.5 to transform the above bound for $\widehat{H}$ into one for $\widehat{G}$,

$$\widehat{G}(w_t) \leq 4c_0 \gamma^{-1} L^2 B_X s \sqrt{\frac{\log(2d/\delta)}{n}}.$$

Since $\|w - v\|_2 \leq 1$ implies $G(w) \leq L^2 B_X^2/2$, standard results from Rademacher complexity imply (e.g. Theorem 26.5 of [28]) that with probability at least $1 - \delta$,

$$G(w_t) \leq \widehat{G}(w_t) + \mathbb{E}_{S \sim \mathcal{D}^n} \mathfrak{R}_S(\ell \circ \sigma \circ \mathcal{G}) + 2L^2 B_X^2 \sqrt{\frac{2\log(4/\delta)}{n}},$$

where $\ell(w; x) = (1/2)(\sigma(w^\top x) - \sigma(v^\top x))^2$ and $\mathcal{G}$ are from Lemma 3.9. For the second term above, Lemma 3.9 and rescaling $\delta$ yields that

$$G(w_t) \leq \frac{2L^3 B_X^2}{\sqrt{n}} + \frac{2L^2 B_X^2 \sqrt{2\log(8/\delta)}}{\sqrt{n}} + \frac{4c_0 \gamma^{-1} L^2 B_X s \sqrt{\log(4d/\delta)}}{\sqrt{n}}.$$

Then Claim 3.4 completes the proof for strictly increasing $\sigma$.

When $\sigma$ is ReLU, the proof follows the same argument given in the proof of Theorem 3.3. Denoting the loss function $\tilde{\ell}(w; x) = (1/2)(\sigma(w^\top x) - \sigma(v^\top x))^2 \sigma'(w^\top x)$, we have

$$\mathbb{E}_{S \sim \mathcal{D}^n} \mathfrak{R}_S \left( \tilde{\ell} \circ \sigma \circ \mathcal{G} \right) \leq \frac{2B_X^2}{\sqrt{n}}. \tag{D.6}$$

By Lemmas D.1 and D.4, there exists some $w_t$, $t < T$ and $\|w_t - v\|_2 \leq 1$, such that

$$\widehat{H}(w_t) \leq 4LK \leq 4c_0 L^2 B_X s \sqrt{\frac{\log(2d/\delta)}{n}}. \tag{D.7}$$

Using standard results from Rademacher complexity,

$$H(w_t) \leq \widehat{H}(w_t) + \mathbb{E}_{S \sim \mathcal{D}^n} \mathfrak{R}_S(\tilde{\ell} \circ \sigma \circ \mathcal{G}) + 2B_X^2 \sqrt{\frac{2\log(4/\delta)}{n}}.$$

By (D.6), this means

$$H(w_t) \leq \frac{4c_0 B_X s \sqrt{\log(4d/\delta)}}{\sqrt{n}} + \frac{2B_X^2}{\sqrt{n}} + \frac{2B_X^2 \sqrt{2\log(8/\delta)}}{\sqrt{n}}.$$

Since $\mathcal{D}$ satisfies Assumption 3.2 and $\|w_t - v\|_2 \leq 1$, Lemma 3.5 shows that $G(w_t) \leq 8\sqrt{2}\alpha^{-4}\beta^{-1} H(w_t)$. Then Claim 3.4 translates the bound for $G(w_t)$ into one for $F(w_t)$. $\qquad \square$

# E   Realizable setting

In this section we assume $y = \sigma(v^\top x)$ a.s. for some $\|v\|_2 \leq 1$. As in the agnostic and noisy teacher network setting, we use the auxiliary loss

$$H(w) := (1/2)\mathbb{E}_{x \sim \mathcal{D}}[(\sigma(w^\top x) - \sigma(v^\top x))^2 \sigma'(w^\top x)].$$

Note that in the realizable setting, the previous auxiliary loss $G$ defined in (3.4) coincides with the true objective $F$, i.e. we have

$$F(w) := (1/2)\mathbb{E}_{x \sim \mathcal{D}}[(\sigma(w^\top x) - \sigma(v^\top x))^2].$$

For purpose of comparison with Yehudai and Shamir [35], we provide analyses for two settings in the realizable case: in the first setting, we consider gradient descent on the population loss,

$$w_{t+1} = w_t - \eta \nabla F(w_t), \tag{E.1}$$

and return $w_{t^*} := \operatorname{argmin}_{0 \leq t < T} F(w_t)$. The second setting is online SGD with samples $x_t \sim \mathcal{D}$. Here we compute unbiased estimates (conditional on $w_t$) of the population risk $F_t(w_t) := (1/2)(\sigma(w_t^\top x_t) - \sigma(v^\top x_t))^2$ and update the weights by

$$w_{t+1} = w_t - \eta \nabla F_t(w_t) \tag{E.2}$$

For SGD, we output $w_{t^*} = \operatorname{argmin}_{0 \leq t < T} F_t(w_t)$.

We summarize our results in the realizable case in Theorem E.1.

**Theorem E.1.** Suppose $\|x\|_2 \leq B$ a.s. and $\sigma$ is non-decreasing and $L$-Lipschitz. Let $\eta \leq L^{-2}B^{-2}$ be the step size.

(a) Let $\sigma$ satisfy Assumption 3.1, and let $\gamma$ be the constant corresponding to $\rho = 4B$. For any initialization satisfying $\|w_0\|_2 \leq 2$, if we run gradient descent on the population risk $T = \lceil 2\varepsilon^{-1}L\eta^{-1}\gamma^{-1}\|w_0 - v\|_2^2 \rceil$ iterations, then there exists $t < T$ such that $F(w_t) \leq \varepsilon$. For stochastic gradient descent, for any $\delta > 0$, running SGD for $\tilde{T} = 6T\log(1/\delta)$ guarantees there exists $w_t$, $t < T$, such that w.p. at least $1 - \delta$, $F(w_t) \leq \varepsilon$.

(b) Let $\sigma$ be ReLU and further assume that $\mathcal{D}$ satisfies Assumption 3.2 for constants $\alpha, \beta > 0$ and that $w_0 = 0$. Let $\nu = \alpha^4\beta/8\sqrt{2}$. If we run gradient descent on the population risk $T = \lceil 2\varepsilon^{-1}L\eta^{-1}\nu^{-1}\|w_0 - v\|_2^2 \rceil$ iterations, then there exists $t < T$ such that $F(w_t) \leq \varepsilon$. For stochastic gradient descent, for any $\delta > 0$, running SGD for $\tilde{T} = 6T\log(1/\delta)$ guarantees there exists $w_t$, $t < T$, such that w.p. at least $1 - \delta$, $F(w_t) \leq \varepsilon$.

A few remarks on the above theorem: first, in comparison with our noisy neuron result in Theorem 4.1, we are able to achieve $\mathsf{OPT} + \varepsilon = \varepsilon$ population risk with sample complexity and runtime of order $\varepsilon^{-1}$ rather than $\varepsilon^{-2}$ using the same assumptions by invoking a martingale Bernstein inequality rather than Hoeffding. Second, although Theorem E.1 requires the distribution to be bounded almost surely, we show in Section E.1 below that for GD on the population loss, we can accomodate essentially any distribution with finite expected squared norm.

Yehudai and Shamir [35] used the marginal spread assumption to show that with probability 1/2, a single neuron in the realizable setting can be learned using gradient-based optimization with random initialization for Lipschitz activation functions satisfying $\inf_{0 < z < \alpha} \sigma'(z) > 0$, where $\alpha$ is the same constant in Assumption 3.2, and thus includes essentially all neural network activation functions like softplus, sigmoid, tanh, and ReLU. Under the additional assumption of spherical symmetry, they showed that this can be improved to a high probability guarantee for the ReLU activation. For gradient descent on the population risk, they proved linear convergence, i.e. a runtime of order $O(\log(1/\varepsilon))$, while for SGD their runtime and sample complexity is of order $O(\varepsilon^{-2}\log(1/\varepsilon))$. In comparison, our result for the non-ReLU activations requires only boundedness of the distributions and holds with high probability over random initializations, with runtime and sample complexity of order $O(\varepsilon^{-1})$ for both gradient descent on the population risk and SGD. Our results for ReLU use the same marginal spread assumption as Yehudai and Shamir, but our proof technique differs in that we do not require the angle $\theta(w_t, v)$ between the weights in the GD trajectory and the target neuron be decreasing. As they pointed out, angle monotonicity fails to hold for the trajectory of gradient descent even when the distribution is a non-centered Gaussian, so that proofs based on angle monotonicity will not translate to more general distributions. Indeed, our proofs in the agnostic and noisy teacher network setting use essentially the same proof technique as the realizable case without relying on angle monotonicity. Instead, we show a type of inductive bias of gradient descent in the sense that if initialized at the origin, the angle between the target vector and the population risk minimizer cannot become larger than $\pi/2$, even in the agnostic setting.

## E.1 Gradient descent on population loss

The key lemma for the proof is as follows.

**Lemma E.2.** Consider gradient descent on the population risk given in (E.1). Let $w_0$ be the initial point of gradient descent and assume $\|w_0\|_2 \leq 2$. Suppose that $\mathcal{D}$ satisfies $\mathbb{E}_x[\|x\|_2^2] \leq B^2$. Let $\sigma$ be non-decreasing and $L$-Lipschitz. Assume the step size satisfies $\eta \leq L^{-2}B^{-2}$. Then for any $T \in \mathbb{N}$, we have for all $t = 0, \ldots, T - 1$, $\|w_t - v\|_2 \leq \|w_0 - v\|_2$, and

$$\|w_0 - v\|_2^2 - \|w_T - v\|_2^2 \geq \eta L^{-1} \sum_{t=0}^{T-1} H(w_t).$$

*Proof.* We begin with the identity, for $t < T$,

$$\|w_t - v\|_2^2 - \|w_{t+1} - v\|_2^2 = 2\eta \langle \nabla F(w_t), w_t - v \rangle - \eta^2 \|\nabla F(w_t)\|_2^2. \qquad (\text{E.3})$$

First, we have

$$\|\nabla F(w_t)\|_2 \leq \mathbb{E}_x \left\| (\sigma(w_t^\top x) - \sigma(v^\top x))\sigma'(w_t^\top x)x \right\|_2$$

$$\leq \sqrt{\mathbb{E}_x \left[ \sigma'(w_t^\top x)(\sigma(w_t^\top x) - \sigma(v^\top x))^2 \right]} \sqrt{\mathbb{E}_x \sigma'(w_t^\top x) \|x\|_2^2}$$

$$\leq B\sqrt{L}\sqrt{\mathbb{E}_x \left[ \sigma'(w_t^\top x)(\sigma(w_t^\top x) - \sigma(v^\top x))^2 \right]}.$$

The first inequality is by Jensen. The second inequality uses that $\sigma'(z) \geq 0$ and Hölder, and the third inequality uses that $\sigma$ is $L$-Lipschitz and that $\mathbb{E}[\|x\|_2^2] \leq B^2$. We therefore have the gradient upper bound

$$\|\nabla F(w_t)\|_2^2 \leq 2B^2 L H(w_t). \tag{E.4}$$

For the inner product term of (E.3), since $\sigma'(z) \geq 0$, we can use Fact C.2 to get

$$\langle \nabla F(w_t), w_t - v \rangle \geq L^{-1} \mathbb{E}_x \left[ \left( \sigma(w_t^\top x) - \sigma(v^\top x) \right)^2 \sigma'(w_t^\top x) \right] = 2L^{-1} H(w_t). \tag{E.5}$$

Putting (E.5) and (E.4) into (E.3), we get

$$\|w_t - v\|_2^2 - \|w_{t+1} - v\|_2^2 \geq 4\eta L^{-1} H(w_t) - 2\eta^2 B^2 L H(w_t) \geq 2\eta L^{-1} H(w_t),$$

where we have used $\eta \leq L^{-2}B^{-2}$. Telescoping the above over $t < T$ gives

$$\|w_0 - v\|_2^2 - \|w_T - v\|_2^2 \geq 2\eta L^{-1} \sum_{t=0}^{T-1} H(w_t).$$

Dividing each side by $\eta T$ shows the desired bound. $\qquad \square$

We will show that if $\sigma$ satisfies Assumption 3.1, then Lemma E.2 allows for a population risk bound for essentially any distribution with $\mathbb{E}[\|x\|_2^2] \leq B^2$. In particular, we consider distributions with finite expected norm squared and the possible types of tail bounds for the norm.

**Assumption E.3.** (a) Bounded distributions: there exists $B > 0$ such that $\|x\|_2 \leq B$ a.s.

(b) Exponential tails: there exist $a_0, C_e > 0$ such that $\mathbb{P}(\|x\|_2^2 \geq a) \leq C_e \exp(-a)$ holds for all $a \geq a_0$.

(c) Polynomial tails: there exist $a_0, C_p > 0$ and $\beta > 1$ such that $\mathbb{P}(\|x\|_2^2 \geq b) \leq C_p a^{-\beta}$ holds for all $a \geq a_0$.

If either (a), (b), or (c) holds, there exists $B > 0$ such that $\mathbb{E} \|x\|_2^2 \leq B^2$. One can verify that for (b), taking $B^2 = 2(a_0 \vee C_e)$ suffices, and for (c), $B^2 = 2(a_0 \vee C_p^{1/\beta}/(1 - \beta))$ suffices. In fact, any distribution that satisfies $\mathbb{E} \|x\|_2^2 < \infty$ cannot have a tail bound of the form $\mathbb{P}(\|x\|_2^2 \geq a) = \Omega(a^{-1})$, since in this case we would have

$$\mathbb{E} \|x\|_2^2 = \int_0^\infty \mathbb{P}(\|x\|_2^2 > t) dt \geq C \int_{a_0}^\infty t^{-1} dt = \infty.$$

So the polynomial tail assumption (c) is tight up to logarithmic factors for distributions with finite $\mathbb{E} \|x\|_2^2$.

**Theorem E.4.** Let $\mathbb{E}[\|x\|_2^2] \leq B^2$ and assume $\mathcal{D}$ satisfies one of the conditions in Assumption E.3. Let $\sigma$ satisfy Assumption 3.1.

(a) Under Assumption E.3a, let $\gamma$ be the constant corresponding to $\rho = 4B$ in Assumption 3.1. Running gradient descent for $T = \lceil 2\varepsilon^{-1} L\eta^{-1}\gamma^{-1} \|w_0 - v\|_2^2 \rceil$ guarantees there exists $t \in [T - 1]$ such that $F(w_t) \leq \varepsilon$.

(b) Under Assumption E.3b, let $\gamma$ be the constant corresponding to $\rho = 4\sqrt{\log(18C_e/\varepsilon)}$. Running gradient descent for $T = \lceil 2\varepsilon^{-1} L\eta^{-1}\gamma^{-1} \|w_0 - v\|_2^2 \rceil$ guarantees there exists $t \in [T - 1]$ such that $F(w_t) \leq \varepsilon$.

(c) Under Assumption E.3c, let $\gamma$ be the constant corresponding to $\rho = 4(18C_p/\varepsilon(\beta-1))^{(1-\beta)/2}$. Running gradient descent for $T = \lceil 2\varepsilon^{-1}L\eta^{-1}\gamma^{-1}\|w_0 - v\|_2^2 \rceil$ guarantees there exists $t \in [T-1]$ such that $F(w_t) \leq \varepsilon$.

*Proof.* First, note that the conditions of Lemma E.2 hold, so that we have for all $t = 0, \dots, T-1$, $\|w_t\|_2 \leq 4$ and

$$\eta \sum_{t=0}^{T-1} H(w_t) \leq L\|w_0 - v\|_2^2 - L\|w_T - v\|_2^2. \tag{E.6}$$

By taking $T = \zeta^{-1}L\varepsilon^{-1}\eta^{-1}\|w_0 - v\|_2^2$ for arbitrary $\zeta > 0$, (E.6) implies that there exists $t \in [T-1]$ such that

$$H(w_t) = \mathbb{E}\left[\left(\sigma(w_t^\top x) - \sigma(v^\top x)\right)^2 \sigma'(w_t^\top x)\right] \leq \frac{L\|w_0 - v\|_2^2}{\eta T} \leq \zeta\varepsilon. \tag{E.7}$$

It therefore suffices to bound $F(w_t)$ in terms of the left hand side of (E.7). We will do so by using the distributional assumptions given in Assumption E.3 and by choosing $\zeta$ appropriately.

We begin by noting that (E.7) implies, for any $\rho > 0$,

$$\mathbb{E}\left[\left(\sigma(w_t^\top x) - \sigma(v^\top x)\right)^2 \sigma'(w_t^\top x)\mathbb{1}(|w_t^\top x| \leq \rho)\right] \leq \zeta\varepsilon. \tag{E.8}$$

For any $\rho > 0$, since $\|w_t\|_2 \leq 4$, the inclusion

$$\left\{\|x\|_2 \leq \rho/4\right\} \subset \left\{|w_t^\top x| \leq \rho\right\}, \tag{E.9}$$

holds. Under Assumption E.3a, by taking $\rho = 4B$ and letting $\gamma$ be the corresponding constant from Assumption 3.1, eqs. (E.8) and (E.9) imply

$$\gamma\mathbb{E}\left[\left(\sigma(w_t^\top x) - \sigma(v^\top x)\right)^2\right] \leq \mathbb{E}\left[\left(\sigma(w_t^\top x) - \sigma(v^\top x)\right)^2 \sigma'(w_t^\top x)\mathbb{1}(\|x\|_2 \leq \rho/4)\right] \leq \zeta\varepsilon.$$

By taking $\zeta = \gamma/2$, this implies $F(w_t) \leq \varepsilon$.

Under Assumption E.3b, by taking $\rho = 4\sqrt{a_0}$, we get

$$\mathbb{E}\left[\|x\|_2^2\mathbb{1}(\|x\|_2^2 > \rho^2/4^2)\right] = \int_{a_0}^{\infty} \mathbb{P}(\|x\|_2^2 > t)dt$$
$$\leq C_e\exp(-a_0). \tag{E.10}$$

Note that Assumption E.3b holds if we take $a_0$ larger. We can therefore let $a_0$ be large enough so that $a_0 \geq \log(18C_e/\varepsilon)$, so that then

$$\mathbb{E}\left[\|x\|_2^2\mathbb{1}(\|x\|_2^2 > \rho^2/4^2)\right] \leq \varepsilon/18. \tag{E.11}$$

Similarly, under Assumption E.3c, we can let $\gamma$ be the constant corresponding to $\rho = 4\sqrt{a_0}$ and take $a_0 \geq (\varepsilon(\beta-1)/18C_p)^{1/(1-\beta)}$ so that

$$\mathbb{E}\left[\|x\|_2^2\mathbb{1}(\|x\|_2^2 > \rho^2/4^2)\right] = \int_{a_0}^{\infty} \mathbb{P}(\|x\|_2^2 > t)dt$$
$$\leq C_p\frac{a_0^{1-\beta}}{\beta-1}$$
$$\leq \varepsilon/18.$$

and so (E.11) holds as well under Assumption E.3c. We can therefore bound

$$\mathbb{E}\left[\left(\sigma(w_t^\top x) - \sigma(v^\top x)\right)^2\mathbb{1}(\|x\|_2^2 > \rho^2/4^2)\right] \leq \mathbb{E}\left[\|w_t - v\|_2^2\|x\|_2^2\mathbb{1}(\|x\|_2^2 > \rho^2/4^2)\right]$$
$$\leq \|w_0 - v\|_2^2\mathbb{E}\left[\|x\|_2^2\mathbb{1}(\|x\|_2^2 > \rho^2/4^2)\right]$$

$$\leq \|w_0 - v\|_2^2 \varepsilon/18$$
$$\leq \varepsilon/2. \tag{E.12}$$

The first inequality uses that $\sigma$ is 1-Lipschitz and Cauchy–Schwarz. The second inequality uses (E.6). The third inequality uses (E.11). The final inequality uses that $\|w_0 - v\|_2 \leq \|w_0\|_2 + \|v\|_2 \leq 3$.

We can then guarantee

$$
\begin{aligned}
2\gamma F(w_t) &= \gamma \mathbb{E}\left[ \left( \sigma(w_t^\top x) - \sigma(v^\top x) \right)^2 \right] \\
&= \mathbb{E}\left[ \left( \sigma(w_t^\top x) - \sigma(v^\top x) \right)^2 \gamma \mathbb{1}(|w_t^\top x| \leq \rho) \right] \\
&\quad + \gamma \mathbb{E}\left[ \left( \sigma(w_t^\top x) - \sigma(v^\top x) \right)^2 \mathbb{1}(|w_t^\top x| > \rho) \right] \\
&\leq \mathbb{E}\left[ \left( \sigma(w_t^\top x) - \sigma(v^\top x) \right)^2 \sigma'(w_t^\top x) \mathbb{1}(|w_t^\top x| \leq \rho) \right] \\
&\quad + \gamma \mathbb{E}\left[ \left( \sigma(w_t^\top x) - \sigma(v^\top x) \right)^2 \mathbb{1}(\|x\|_2^2 > \rho^2/4^2) \right] \\
&\leq \zeta\varepsilon + \gamma\varepsilon/2 \\
&\leq \gamma\varepsilon.
\end{aligned}
$$

The first inequality follows since Assumption 3.1 implies $\sigma'(z)\mathbb{1}(|z| \leq \rho) \geq \gamma\mathbb{1}(|z| \leq \rho)$ and by (E.9). The second inequality uses (E.8) and (E.12). The final inequality takes $\zeta = \gamma/2$. □

**Remark E.5.** The precise runtime guarantee in Theorem E.1 will depend upon the activation function and tail distribution. The worst-case activation functions (like the sigmoid) can have $\gamma \sim \exp(-\rho)$, and so if one only has polynomial tails, the runtime can be exponential in $\varepsilon^{-1}$ in this case. If the distribution of $\|x\|_2^2$ has exponential tails, as is the case if the components of $x$ are sub-Gaussian, runtime will be polynomial in $\varepsilon^{-1}$. On the other hand, if the $\gamma$ in Assumption 3.1 is a fixed constant independent of $\rho$ (as it is for the leaky ReLU), any of the tail bounds under consideration will have runtime of order $\varepsilon^{-1}$.

## E.2 Stochastic gradient descent proofs

We consider the online version of stochastic gradient descent, where we sample independent samples $x_t \sim \mathcal{D}$ at each step and compute stochastic gradient updates $g_t$, such that

$$g_t = \left( \sigma(w_t^\top x_t) - \sigma(v^\top x_t) \right) \sigma'(w_t^\top x_t) x_t, \quad w_{t+1} = w_t - \eta g_t.$$

As in the gradient descent case, we have a key lemma that relates the distance of the weights at iteration $t$ from the optimal $v$ with the distance from initialization and the cumulative loss.

**Lemma E.6.** Assume that $\sigma$ is non-decreasing and $L$-Lipschitz, and that $\mathcal{D}$ satisfies $\|x\|_2 \leq B$ a.s. Assume the initialization satisfies $\|w_0\|_2 \leq 2$. Let $T \in \mathbb{N}$ and run stochastic gradient descent for $T-1$ iterations at a fixed learning rate $\eta$ satisfying $\eta \leq L^{-2}B^{-2}$. Then with probability one over $\mathcal{D}$, we have $\|w_{t+1} - v\|_2 \leq \|w_t - v\|_2$ for all $t < T$, and

$$\|w_0 - v\|_2^2 - \|w_T - v\|_2^2 \geq 2\eta L^{-1} \sum_{t=0}^{T-1} H_t,$$

where $H_t := \frac{1}{2} \left( \sigma(w_t^\top x_t) - \sigma(v^\top x_t) \right)^2 \sigma'(w_t^\top x_t)$.

*Proof.* We begin with the decomposition

$$\|w_t - v\|_2^2 - \|w_{t+1} - v\|_2^2 = 2\eta \langle g_t, w_t - v \rangle - \eta^2 \|g_t\|_2^2. \tag{E.13}$$

By Assumption 3.1, since $\|x\|_2 \leq B$ a.s. it holds with probability one that

$$\|g_t\|_2^2 = \left\| \left( \sigma(w_t^\top x_t) - \sigma(v^\top x_t) \right) \sigma'(w_t^\top x_t) x_t \right\|_2^2 \leq 2LB^2 H_t. \tag{E.14}$$

By Fact C.2, since $\sigma'(z) \geq 0$, we have with probability one,

$$\langle g_t, w_t - v \rangle = \left( \sigma(w_t^\top x_t) - \sigma(v^\top x_t) \right) \sigma'(w_t^\top x_t)(w_t^\top x_t - v^\top x_t)$$

$$\geq L^{-1}\left(\sigma(w_t^\top x_t) - \sigma(v^\top x_t)\right)^2 \sigma'(w_t^\top x_t)$$
$$= 2L^{-1}H_t. \tag{E.15}$$

Putting (E.14) and (E.15) into (E.13), we get

$$\|w_t - v\|_2^2 - \|w_{t+1} - v\|_2^2 \geq 4\eta L^{-1}H_t - 2\eta^2 LB^2 H_t$$
$$\geq 2\eta L^{-1}H_t,$$

by taking $\eta \leq L^{-2}B^{-2}$. Telescoping over $t < T$ gives the desired bound.

$\square$

We now want to translate the bound on the empirical error to that of the true error. For this we use a martingale Bernstein inequality of Beygelzimer et al. [5]. A similar analysis of SGD was used by Ji and Telgarsky [18] for a one-hidden-layer ReLU network.

**Lemma E.7** (Beygelzimer et al. [5], Theorem 1). *Let $\{Y_t\}$ be a martingale adapted to the filtration $\mathcal{F}_t$, and let $Y_0 = 0$. Let $\{D_t\}$ be the corresponding martingale difference sequence. Define the sequence of conditional variance*

$$V_t := \sum_{k=1}^{t} \mathbb{E}[D_k^2 | \mathcal{F}_{k-1}],$$

*and assume that $D_t \leq R$ almost surely. Then for any $\delta \in (0,1)$, with probability greater than $1 - \delta$,*

$$Y_t \leq R\log(1/\delta) + (e-2)V_t/R.$$

**Lemma E.8.** *Suppose that $\|x\|_2 \leq B$ a.s., and let $\sigma$ be non-decreasing and $L$-Lipschitz. Assume that the trajectory of SGD satisfies $\|w_t - v\|_2 \leq \|w_0 - v\|_2$ for all $t$ a.s. We then have with probability at least $1 - \delta$,*

$$\frac{1}{T}\sum_{t=0}^{T-1} H(w_t) \leq \frac{4}{T}\sum_{t=0}^{T-1} H_t + \frac{2}{T}B^2 L^3 \|w_0 - v\|_2^2 \log(1/\delta).$$

*Proof.* Let $\mathcal{F}_t = \sigma(x_0, \ldots, x_t)$ be the $\sigma$-algebra generated by the first $t + 1$ draws from $\mathcal{D}$. Then the random variable $G_t := \sum_{\tau=0}^{t}(H(w_\tau) - H_\tau)$ is a martingale with respect to the filtration $\mathcal{F}_t$ with martingale difference sequence $D_t := H(w_t) - H_t$. We need bounds on $D_t$ and on $\mathbb{E}[D_t^2|\mathcal{F}_{t-1}]$ in order to apply Lemma E.7.

Since $\sigma$ is $L$-Lipschitz and $\|x\|_2 \leq B$ a.s., with probability one we have

$$D_t \leq H(w_t) \leq \frac{1}{2}L^3 B^2 \|w_t - v\|_2^2 \leq \frac{1}{2}L^3 B^2 \|w_0 - v\|_2^2. \tag{E.16}$$

The last inequality uses the assumption that $\|w_t - v\|_2 \leq \|w_0 - v\|_2$ a.s. Similarly,

$$\mathbb{E}[H_t^2|\mathcal{F}_{t-1}] = \frac{1}{4}\mathbb{E}\left[\left(\sigma(w_t^\top x_t) - \sigma(v^\top x_t)\right)^4 \sigma'(w_t^\top x_t)^2 | \mathcal{F}_{t-1}\right]$$
$$\leq \frac{1}{4}L^3 B^2 \|w_t - v\|_2^2 \mathbb{E}_x\left[\left(\sigma(w_t x_t) - \sigma(v^\top x_t)\right)^2 \sigma'(w_t^\top x_t)|\mathcal{F}_{t-1}\right]$$
$$\leq \frac{1}{2}L^3 B^2 \|w_0 - v\|_2^2 H(w_t). \tag{E.17}$$

In the first inequality, we have used $\|x\|_2^2 \leq B^2$ a.s. and $L$-Lipschitzness of $\sigma$. For the second, we use the assumption that $\|w_t - v\|_2 \leq \|w_0 - v\|_2$ together with the fact that $\mathbb{E}_x[H_t|\mathcal{F}_{t-1}] = H(w_t)$. We then can use (E.17) to bound the squared increments,

$$\mathbb{E}[D_t^2|\mathcal{F}_{t-1}] = H(w_t)^2 - 2H(w_t)\mathbb{E}[H_t|\mathcal{F}_{t-1}] + \mathbb{E}[H_t^2|\mathcal{F}_{t-1}]$$
$$= -H(w_t)^2 + \mathbb{E}[H_t^2|\mathcal{F}_{t-1}]$$
$$\leq \frac{1}{2}L^3 B^2 \|w_0 - v\|_2^2 H(w_t). \tag{E.18}$$

This allows for us to bound

$$V_T := \sum_{t=0}^{T-1} \mathbb{E}[D_t^2|\mathcal{F}_{t-1}] \leq \frac{1}{2}B^2L^3 \left\| w_0 - v \right\|_2^2 \sum_{t=0}^{T-1} H(w_t).$$

Since $D_t \leq H(w_t) \leq (1/2)L^3B^2 \left\| w_0 - v \right\|_2^2$ a.s. by (E.16), Lemma E.7 implies that with probability at least $1 - \delta$, we have

$$\sum_{t=0}^{T-1} (H(w_t) - H_t) \leq (\exp(1) - 2) \sum_{t=0}^{T-1} H(w_t) + \frac{1}{2}L^3B^2 \left\| w_0 - v \right\|_2^2 \log(1/\delta),$$

and using that $(1 - \exp(1) + 2)^{-1} \leq 4$, we divide each side by $T$ and get

$$\frac{1}{T} \sum_{t=0}^{T-1} H(w_t) \leq \frac{4}{T} \sum_{t=0}^{T-1} H_t + \frac{2}{T}L^3B^2 \left\| w_0 - v \right\|_2^2 \log(1/\delta). \tag{E.19}$$

$\square$

With the above in hand, we can prove Theorem E.1 in the SGD setting.

*Proof of Theorem E.1, SGD.* By the assumptions in the theorem, Lemma E.6 holds, so that we have for any $t = 0, \dots, T-1$, $\left\| w_t \right\|_2 \leq 4$ and

$$\left\| w_t - v \right\|_2^2 + 2\eta L^{-1} \sum_{\tau=0}^{t-1} H_\tau \leq \left\| w_0 - v \right\|_2^2. \tag{E.20}$$

This shows that $\left\| w_t - v \right\|_2 \leq \left\| w_0 - v \right\|_2$ holds for all $t = 0, \dots, T-1$ a.s., allowing for the application of Lemma E.8 to get

$$\frac{1}{T} \sum_{t=0}^{T-1} H(w_t) \leq \frac{4}{T} \sum_{t=1}^{T} H_t + \frac{2}{T}L^3B^2 \left\| w_0 - v \right\|_2^2 \log(1/\delta). \tag{E.21}$$

Dividing both sides of (E.20) by $\eta T L^{-1}$ yields

$$\min_{t<T} H(w_t) \leq \frac{1}{T} \sum_{t=0}^{T-1} H(w_t) \leq \frac{L \left\| w_0 - v \right\|_2^2}{\eta T} + \frac{2}{T}L^3B^2 \left\| w_0 - v \right\|_2^2 \log(1/\delta).$$

For arbitrary $\zeta > 0$, taking $T = \lceil 2\varepsilon^{-1}\zeta^{-1}\eta^{-1}L^3B^2 \left\| w_0 - v \right\|_2^2 \log(1/\delta) \rceil$ shows there exists $T$ such that $H(w_t) \leq \zeta\varepsilon$. When $\sigma$ satisfies Assumption 3.1, since $\left\| w_t \right\|_2 \leq 4$ for all $t$, it holds that $H(w_t) \geq \gamma F(w_t)$, so that $\zeta = \gamma$ furnishes the desired bound.

When $\sigma$ is ReLU and $\mathcal{D}$ satisfies Assumption 3.2, we note that Lemma E.6 implies $\left\| w_t - v \right\|_2 \leq \left\| w_0 - v \right\|_2$ a.s. Thus taking $\zeta = \alpha^4\beta/8\sqrt{2}$ and using Lemma 3.5 completes the proof. $\square$

# F   Remaining Proofs

*Proof of Lemma 3.8.* Since $\sigma$ is non-decreasing, $|\sigma(v^\top x) - y| \leq |\sigma(B_X)| + B_Y$. In particular, each summand defining $\widehat{F}(v)$ is a random variable with absolute value at most $a = (|\sigma(B_X)| + B_Y)^2$. As $\mathbb{E}[\widehat{F}(v)] = F(v) = \mathsf{OPT}$, Hoeffding's inequality implies the lemma. $\square$

*Proof of Lemma 3.9.* The bound $\mathfrak{R}_S(\mathcal{G}) \leq 2\max_i \left\| x_i \right\|_2 / \sqrt{n}$ follows since $\left\| w \right\|_2 \leq 2$ holds on $\mathcal{G}$ with standard results Rademacher complexity theory (e.g. Sec. 26.2 of [28]); this shows $\mathfrak{R}(\mathcal{G}) \leq 2B_X/\sqrt{n}$. Using the contraction property of the Rademacher complexity, this implies $\mathfrak{R}(\sigma \circ \mathcal{G}) \leq 2B_X L/\sqrt{n}$. Finally, note that if $\left\| w - v \right\|_2 \leq 1$ and $\left\| x \right\|_2 \leq B_X$, we have

$$\left\| \nabla \ell(w; x) \right\| = \left\| \left( \sigma(w^\top x) - \sigma(v^\top x) \right) \sigma'(w^\top x)x \right\| \leq L^2 \left\| w - v \right\| \left\| x \right\| \leq L^2 B_X. \tag{F.1}$$

In particular, $\ell$ is $L^2 B_X$ Lipschitz. The result follows. $\square$

## Footnotes

[3]Agnostic learning results typically require i.i.d. samples, and adversarial noise may depend on other samples in malicious ways. Even in the i.i.d. case, trouble arises if one wishes to use parameter recovery to show that a given algorithm competes with the population risk minimizer. Consider the ReLU with labels given by $y = \sigma(v^\top x) + \xi$ where $\xi = -\sigma(v^\top x)$. The zero vector minimizes the population risk, and so any algorithm that returns the target neuron $\sigma(v^\top x)$ has large population risk. A similar phenomenon occurs for $\xi = \sigma(v^\top x)$.

[4]Although their result is stated for the ReLU and isotropic log-concave distributions, their results also apply for $L$-Lipschitz activations satisfying $\inf_z \sigma'(z) \geq \gamma > 0$ for isotropic distributions that satisfy our Assumption 3.2. In this setting, one can show that the Chow parameters satisfy $\|\chi(\sigma_u) - \chi(\sigma_w)\|^2 \geq \gamma L^{-1}\mathbb{E}[(\sigma(u^\top x) - \sigma(v^\top x))^2]$, from which the result follows easily.