[Reviews · NeurIPS 2020]

Review 1

Summary and Contributions: The paper proves convergence of gradient descent to population risk of O(OPT^{1/2}) + epsilon for learning a single neuron with some reasonable assumptions on the activation function. This generalizes previous results such as Isotron and GLMtron to the full agnostic case and to the vanilla gradient descent algorithm. The main trick is to bound the objective by auxiliary functions.

Strengths: Strong paper. Significantly improves our understanding of a long standing problem, with a lot of previous work.

Weaknesses: none.

Correctness: I didn't verify all of the proofs but only some of them and the main proof sketch. Seems correct to me.

Clarity: very well written.

Relation to Prior Work: clearly discussed.

Reproducibility: Yes

Additional Feedback:


Review 2

Summary and Contributions: The paper considers the problem of agnostically learning a single neuron with respect to the squared loss via gradient descent (GD). The focus of the paper is on specifically understanding the guarantees GD obtains. Under only boundedness of the input distribution, the authors show that for strictly increasing (gradient bounded below by a constant) activation functions, GD finds a point that achieves error O(\sqrt{opt}) + \eps where opt is the loss of the best fitting neuron. They extend the result to ReLU under standard anti-concentration assumptions (similar to [1,2]). For ReLU, it is known that you in fact get O(opt) + \eps under essentially the same assumptions (see [1]) using an alternate algorithm. Further they show that under the mean-0 sub-gaussian noise setting, the results can be improved to opt + eps. Note that it is known that in this setting there is an algorithm (GLMtron [3]) which gives the same guarantee under weaker assumptions. [1] Diakonikolas, I., Goel, S., Karmalkar, S., Klivans, A.R. and Soltanolkotabi, M., 2020, January. Approximation Schemes for ReLU Regression. In Conference on Learning Theory. [2] Diakonikolas, I., Kontonis, V., Tzamos, C. and Zarifis, N., 2020. Learning Halfspaces with Massart Noise Under Structured Distributions. In Conference on Learning Theory. [3] Kakade, S.M., Kanade, V., Shamir, O. and Kalai, A., 2011. Efficient learning of generalized linear and single index models with isotonic regression. In Advances in Neural Information Processing Systems (pp. 927-935).

Strengths: The paper furthers the study of learning guarantees for vanilla gradient descent in the context of neural networks. Prior to this work, the guarantees known for GD were mostly restricted to the realizable setting and not for the challenging agnostic setting. Despite the fact that the results are obtainable by other algorithms [1,3], I think it is good to know the guarantees obtained via GD since it is the primary choice of algorithm for training NNs.

Weaknesses: The bounds obtained by the authors for GD are not as good as the ones known for prior work especially in the ReLU setup. The paper does not have a lower bound showing that their results are tight. Thus, it is not clear if it is due to some looseness in the analysis in the paper or due to a more fundamental issue. It would have been great if the authors explored this direction a bit further.

Correctness: The proofs seem correct to me. I have checked the proofs in the main text but have not verified all the proofs in the appendix, so it is possible I might have missed some small detail.

Clarity: The paper is generally well-written and easy to follow.

Relation to Prior Work: The authors do a commendable job of putting their work in context with prior known results. They have consolidated the known results for learning a single neuron under varying noise models as well as distributional assumptions. I believe this is useful for anyone hoping to work in this area. I also appreciate the honesty of mentioning when prior work gets stronger guarantees. One important work that the authors have missed is [1]. Since this work is especially relevant to the paper, the authors should add relevant comparison/discussion.

Reproducibility: Yes

Additional Feedback: Minor typos: - In equation (3.3) and (3.4) ā€˜vā€™ is not defined which makes it a bit confusing. - Also proof of Claim 3.4 is missing. [Post Author Response] My score remains unchanged. The bounds in the paper are not optimal however I still think the work makes good progress towards our understanding of GD. I believe the results from Diakonikolas et. al. can be extended to the strictly increasing setting in a straightforward manner. I do agree that the authors' result is under more general distributional setting however achieves a weaker guarantee. As far as the dimension dependence goes, I think this is an artifact of the norm of the input (the paper assumes norm 1 whereas for gaussian it would be sqrt{d}).


Review 3

Summary and Contributions: This paper considers learning a single neuron using gradient descent. If the optimal solution achieves a population risk of OPT, then gradient descent can roughly achieve a population risk of sqrt{OPT} if there is no relationship between features and labels, and OPT in the noisy teacher network setting.

Strengths: This paper provides a thorough solution to the problem of learning a single neuron with gradient descent. The proofs are clean, and contain some interesting ideas: for example, that the gradient points to a good direction before finding a low risk.

Weaknesses: The main limitation is that only one neuron is allowed. It would be more interesting if the analysis can handle more complicated models, such as two-layer networks.

Correctness: I do not see any correctness issue.

Clarity: I think the paper is well-written.

Relation to Prior Work: The discussion of prior work is thorough as far as I know.

Reproducibility: Yes

Additional Feedback: Reply to the feedback: Thanks for the response! I agree that agnostic learning of a single neuron is an interesting and important problem, and this paper provides a thorough analysis of it. On the other hand, while I agree it is natural to start from the single neuron case, I am not sure whether techniques in this setting can help us analyze the multiple neuron case: for example, can we generalize eq. (3.6) to the general case? It would be better to include some discussion.


Review 4

Summary and Contributions: The paper analyzes the complexity of learning with a single neuron by gradient descent both in agnostic setting and noisy teacher network setting. The main contribution is being a first result for agnostic learning of a single neuron.

Strengths: The paper is very clearly written and has novel result in the agnostic learning setting.

Weaknesses: The significance is a bit limited by restricting to a single neuron setting since much of the difficulty of learning a neural network lies in jointly learning the mixture weights with the neurons.

Correctness: Yes.

Clarity: Yes.

Relation to Prior Work: Yes.

Reproducibility: Yes

Additional Feedback: In general, a very well-written paper that is very clear in introducing the problems as well as explaining the main proof strategy. The main contribution is in the agnostic learning setting since the noisy teacher setting is very similar to GLM. ========================================== After reading other reviews and author's rebuttal, I maintain the same rating.

[Author Response · NeurIPS 2020]

We thank all the reviewers for their helpful comments. We address specific questions below.

**Reply to Reviewer 2**

Question: New work by Diakonikolas et al.

We thank the reviewer for their thorough review and for alerting us to the recent work by Diakonikolas et al. [1].
We will be sure to provide a detailed comparison with this paper in the camera-ready. [1] showed that learning the
single ReLU neuron up to $O(\mathsf{OPT}) + \varepsilon$ risk for log-concave and isotropic distributions is possible if one uses gradient
descent on a convex surrogate risk for the squared loss; it was previously known that learning up to exactly $\mathsf{OPT} + \varepsilon$
is impossible in polynomial time [3]. The updates by gradient descent on this surrogate correspond to the GLMTron
updates of [2]. By contrast, our bounds for strictly increasing and Lipschitz activations cover *any* distribution over $x$
with bounded marginals, with dimension-independent sample complexity, by minimizing the (nonconvex) empirical
risk with vanilla G.D., although we achieve a weaker guarantee of $O(\mathsf{OPT}^{1/2})$. Although our risk guarantee is weaker,
we believe a complete characterization of what distributions can be agnostic PAC learned using neural networks trained
by gradient descent on the empirical risk is a fundamental research question. Our work provides, to our knowledge, the
first positive result on this question for the single neuron in the agnostic PAC learning setting.

Question: Lower bounds

Thank you for your suggestion for studying lower bounds for this problem. There are two types of lower bounds that we
are interested in: (1) whether there exist distributions for which no algorithm can achieve population risk $\mathsf{OPT} + \varepsilon$ for
the single neuron; (2) whether there exist distributions for which gradient descent on the empirical risk cannot achieve
population risk $\mathsf{OPT} + \varepsilon$ (or even $O(\mathsf{OPT}^{1/2}) + \varepsilon$). For ReLU, [3] addresses (1), and [4] addresses (2), but to our
knowledge no such results are known for nontrivial strictly increasing and Lipschitz activations. We hope to explore
these questions in future work.

**Reply to Reviewer 3 and Reviewer 4**

Question: Significance of agnostic learning of a single neuron

We thank the reviewers for their comments. We agree with the reviewers that the agnostic PAC learning of neural
networks with multiple neurons and layers is an important problem. Unfortunately, there are very few works that have
been able to show any results in this direction, as we describe in lines 41–53 and 75–92 of our submission. Even in the
single neuron setting, the question of what distributions can be PAC learned has only recently begun to be understood
[1,3,4].

We believe that the agnostic PAC learning of the single neuron using gradient descent is a fundamental problem for
the understanding of neural networks. Without a full characterization of what can be learned for the simplest possible
neural network—the single neuron—it seems unlikely that we will find satisfying explanations for why complicated,
deep neural networks trained by gradient descent are so successful. As our work is the first result for agnostic PAC
learning for a single neuron using gradient descent on the empirical risk, we think we have made significant progress on
this problem.

[1] I. Diakonikolas, S. Goel, S. Karmalkar, A. R. Klivans, and M. Soltanolkotabi. Approximation schemes for relu
regression. In *Conference on Learning Theory (COLT)*, 2020.

[2] S. M. Kakade, A. Kalai, V. Kanade, and O. Shamir. Efficient learning of generalized linear and single index models
with isotonic regression. In *Conference on Neural Information Processing Systems (NeurIPS)*, 2011.

[3] S. Goel, S. Karmalkar, and A. R. Klivans. Time/accuracy tradeoffs for learning a relu with respect to gaussian
marginals. In *Conference on Neural Information Processing Systems (NeurIPS)*, 2019.

[4] G. Yehudai and O. Shamir. Learning a single neuron with gradient methods. In *Conference on Learning Theory
(COLT)*, 2020.


[Meta-Review · NeurIPS 2020]

The reviewers agree that the techniques in this paper are interesting and novel, and that the paper is well-written. For the camera ready version, the authors should include discussion for the additional references pointed out by Reviewer #2. In particular, it would be good to point out that one of the main strengths of this work is the fact that it handles arbitrary distributions over x. Per the comments of Reviewers 3 and 4, it would also be nice to include some additional discussion of the challenges of extending these techniques to multiple neurons.